# CMMA: Benchmarking Multi-Affection Detection in Chinese Multi-Modal Conversations

**Yazhou Zhang**
The Hong Kong Polytechnic University,
China Mobile Communication Group Tianjin Co.,
Zhengzhou University of Light Industry
yzhou_zhang@tju.edu.cn

**Yang Yu**
Zhengzhou University of Light Industry
yuyang19980818@outlook.com

**Qing Guo**
IHPC and CFAR, Agency for Science,
Technology and Research, Singapore
tsingqguo@ieee.org

**Benyou Wang**
The Chinese University of Hong Kong, Shenzhen,
Shenzhen Research Institute of Big Data
wabyking@gmail.com

**Dongming Zhao**
China Mobile Communication Group Tianjin Co.
waitman_840602@163.com

**Sagar Uprety**
Bravura Solutions
suprety@bravurasolutions.com

**Dawei Song**
School of Computer Science and Technology, Beijing Institute of Technology
dwsong@bit.edu.cn

**Qiuchi Li**[*]
Copenhagen University
qiuchi.li@di.ku.dk

**Jing Qin**
The Hong Kong Polytechnic University
harry.qin@polyu.edu.hk

## Abstract

Human communication has a multi-modal and multi-affect nature. The inter-relatedness of different emotions and sentiments poses a challenge to jointly detect multiple human affects with multi-modal clues. Recent advances in this field employed multi-task learning paradigms to render the inter-relatedness across tasks, but the scarcity of publicly available resources sets a limit to the potential of works. To fill this gap, we build the first Chinese Multi-modal Multi-Affect conversation (CMMA) dataset, which contains 3,000 multi-party conversations and 21,795 multi-modal utterances collected from various styles of TV-series. CMMA contains a wide variety of affect labels, including sentiment, emotion, sarcasm and humor, as well as the novel inter-correlations values between certain pairs of tasks. Moreover, it provides the topic and speaker information in conversations, which promotes better modeling of conversational context. On the dataset, we empirically analyze the influence of different data modalities and conversational contexts on different affect analysis tasks, and exhibit the practical benefit of inter-task correlations. The full dataset will be publicly available for research[2].

---

[*]Corresponding author: Qiuchi Li

[2]https://github.com/annoymity2022/Chinese-Dataset

37th Conference on Neural Information Processing Systems (NeurIPS 2023) Track on Datasets and Benchmarks.

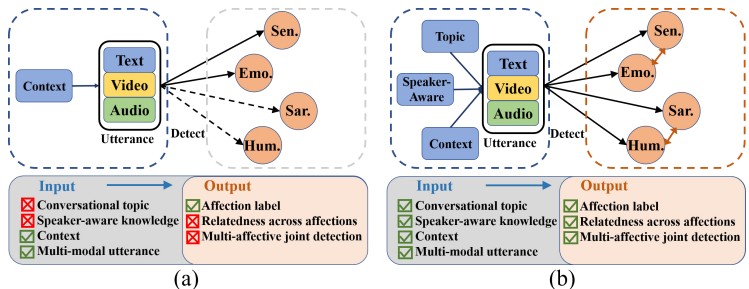

Figure 1: Previous works neglect topic, speaker knowledge and relatedness across affects (a). Our work can incorporate all of them into an unified framework (b). Abbreviations: Sen.: Sentiment, Emo.: Emotion, Sar.: Sarcasm, Hum.: Humor.

# 1 Introduction

Human communication is multi-modal with textual (e.g., natural language), visual (e.g., facial gestures) and audio (e.g., tones) channels, and also multi-affective in that different types of affects, such as sentiment, sarcasm and humor, are often expressed in a mixed manner. The interactions of different modalities and inter-correlations between different affect types bring opportunities as well as challenges for multi-modal affect detection, especially in a conversational context. Research works traditionally targeted at sentiment [1] and emotion [2], the two most common affect types. More recently, increasing attention has been placed on sarcasm detection [3] and humor detection [4].

To set the scope, we target the above four affect detection tasks, which are inherently inter-related. Sentiment and emotion correspond to long-term and short-term human subjective experience and mental attitude, where stable sentiments are rooted in and expressed through emotion. Sarcasm is a subtle form of human figurative language that intends to express criticism, humor or anger emotions, while humor is treated as a sentimental tendency to provoke laughter and provide amusement, often in a sarcastic way [5]. The recognition of one affect type may potentially benefit the understanding of the other.

*Related work.* Despite the crucial role of inter-relatedness between different affect types, they were not sufficiently captured in existing multi-modal affect detection works, largely due to the lack of publicly available datasets for this purpose. Traditional works simply rendered the tasks as being mutually independent and ignored the mutual influence across them [6]. The recent popularity of the multi-task learning paradigm led to an emerging focus on multi-affect joint detection, such as sentiment-emotion joint recognition [7], sentiment-sarcasm joint classification [3], and sarcasm-humor joint detection [8]. However, the datasets they worked on, such as MELD [9], IEMOCAP [10], UR-FUNNY [5], MUStARD [11], etc., have annotations on solely one or two types of affect, and inter-relatedness between tasks is absent. Without an explicit annotation of cross-task correlations, the potential of multi-modal multi-affect joint detection could not be fully explored, neither deepen the understanding on human complicated affects.

We fill the gap by constructing a large-scale benchmark multi-modal multi-affect conversational dataset. We manage to tackle the following main challenges for building such a dataset: (1) *multi-affect joint judgment*: the subjectivity and creativity of human language make it hard to judge different affects at the same time accurately; (2) *multi-affect correlation*: different affects can be indistinguishable at certain circumstances, and it is difficult to accurately measure their relatedness; (3) *context effect*: an utterance may express different affects in different conversational contexts.

To address above challenges, we build a novel **C**hinese **M**ulti-modal **M**ulti-**A**ffection conversation dataset, termed CMMA. It consists of 21,795 multi-modal utterances from 3,000 multi-party conversations collected from various TV series with many kinds of subjects and styles, e.g., comedy, thriller, drama, etc. Each utterance is manually annotated with sentiment (including pride and romantic love), emotion, sarcasm and humor labels, accompanied by sentiment-emotion and sarcasm-humor inter-relatedness measures. Considering that the external knowledge implicitly influences the speaker's affective state, the speaker's background (i.e., name, profession, sex, personality) and the topic of each conversation are provided (c.f. Fig.1 in supplementary document). CMMA supports both single-task and multi-task learning paradigms for human affect understanding. More importantly,

Table 1: Comparison of CMMA with other datasets. ✓ denotes that the dataset only provides the speaker's name.

| Dataset | Type | Size | Modality | Resource | Language | Annotation | Inter-Task Correlation | Speaker Information | Topic |
|---|---|---|---|---|---|---|---|---|---|
| YouTube | Video | 47 | Text, Image, Speech | YouTube | English | Sentiment | ✗ | ✗ | ✗ |
| MOUD | Video | 498 | Text, Image, Speech | YouTube | English | Sentiment | ✗ | ✗ | ✗ |
| MOSI | Video | 2,199 | Text, Image, Speech | YouTube | English | Sentiment | ✗ | ✗ | ✗ |
| CH-SIMS | Video | 2,281 | Text, Image, Speech | Movie, TV | Chinese | Sentiment | ✗ | ✗ | ✗ |
| IEMOCAP | Dialogue | 10,039 | Text, Image, Speech | Performance | English | Emotion | ✗ | ✓ | ✗ |
| MELD | Dialogue | 13,708 | Text, Image, Speech | TV Show | English | Sentiment, Emotion | ✗ | ✓ | ✗ |
| MEISD | Dialogue | 20,000 | Text, Image, Speech | TV Show | English | Sentiment, Emotion | ✗ | ✗ | ✗ |
| ScenarioSA | Dialogue | 24,072 | Text | Social Media | English | Sentiment | ✗ | ✗ | ✓ |
| MUStARD | Dialogue | 690 | Text, Image, Speech | TV Show | English | Sarcasm | ✗ | ✓ | ✗ |
| Twitter | Tweet | 24,635 | Text, Image | TV Show | English | Sarcasm | ✗ | ✓ | ✗ |
| Silver-Standard | Instagram post | 20K | Text, Image, Speech | TV Show | English | Sarcasm | ✗ | ✓ | ✗ |
| MHD | Dialogue | 13,633 | Text, Image, Speech | TV Show | English | Humor | ✗ | ✓ | ✗ |
| BBT | Dialogue | 39,769 | Text, Image, Speech | TV Show | English | Humor | ✗ | ✓ | ✗ |
| UR-FUNNY | TED talk | 16,514 | Text, Image, Speech | TV Show | English | Humor | ✗ | ✓ | ✓ |
| MUMOR | Dialogue | 19,103 | Text, Image, Speech | TV Show | English, Chinese | Sentiment, Emotion, Humor | ✗ | ✓ | ✗ |
| MaSaC | Dialogue | 15,000 | Text, Image, Speech | TV Show | English,Hindi | Sarcasm, Humor | ✗ | ✓ | ✗ |
| Memotion | Internet Meme | 8,871 | Text, Image | Social Media | English | Sentiment, Emotion, Sarcasm, Humor, Offensive, Motivational | ✗ | ✗ | ✗ |
| **CMMA (Ours)** | **Dialogue** | **21,795** | **Text, Image, Speech** | **TV Show** | **Chinese** | **Sentiment, Emotion, Sarcasm, Humor, Pride, Love** | ✓ | ✓ | ✓ |

as illustrated in Figure 1, CMMA can support multi-task learning paradigms better than existing datasets, with richer multi-task , multi-modal clues and more external knowledge. We present an overview of related resources covering sentiment, emotion, sarcasm and humor with a comprehensive comparison in Table 1. Please refer to App.B (in supplementary document) for more details.

We further conduct a pilot study on the CMMA dataset by employing state-of-the-art (SOTA) multi-modal models for a small range of tasks with all combinations of input features. We find out that multi-affect detection is more dependent on conversational context, but the speaker and topic information is also beneficial. In addition, we empirically show the benefits of sentiment-emotion and human-sarcasm inter-relatedness to multi-task joint detection frameworks. In summary, our major contributions are presented as follows:

- We construct the first Chinese multi-modal multi-affect conversation dataset annotated with the sentiment, emotion, sarcasm, and humor labels. It provides a benchmark for multi-affect detection frameworks.

- We make the first attempt to manually annotate the relevance intensity between sentiment and emotion, and between sarcasm and humor. This will help determine the main-secondary task and improve current multi-task learning frameworks. In addition, the dataset is the first of its kind that includes the speaker's information (i.e., name, profession, sex, personality) and conversation topics.

- We provide the annotations of sentiment, emotion, sarcasm and humor, along with well-illustrated quality control and agreement analysis.

- We show a comprehensive statistics of the dataset, covering the distribution of TV sources, characters and affect types.

- We propose six multi-modal affect detection tasks to evaluate CMMA. The results of SOTA baselines using different feature combinations suggest the need for multi-task learning models.

## 2 CMMA Dataset

### 2.1 Data Acquisition

We aim to create a large-scale multi-modal multi-affect dataset to support affect understanding. Following the rule of "representativeness" in corpus linguistics' guidelines [12], the samples is diverse across (1) *domains:* a broad range of domains should be included to cover the expressions of sentiment, emotion, sarcasm, humor, pride and love; (2) *speakers:* different speakers have various means of expressing affect; (3) *topics:* topics may have sentimental tendencies. For example, crimes and disasters often indicate negative sentiment, while romance and love suggest the opposite; (4) *modalities:* affect communication is multi-modal, covering natural language (text), facial expression (vision) and acoustic tonality (audio).

Table 2: Statistics of CMMA. (t,v,a) = (text, video, audio).

| Item | Train | Dev | Test |
|---|---|---|---|
| #Modalities | (t,v,a) | (t,v,a) | (t,v,a) |
| #Conversations | 1800 | 600 | 600 |
| #Utterances | 13788 | 4046 | 3961 |
| #Speakers | 299 | 78 | 119 |
| #Words | 115,434 | 35,487 | 34,521 |
| #Unique words | 2,677 | 1,842 | 1,988 |
| #Video duration | 9.2h | 3.0h | 3.0h |
| #Average utterances per conversation | 7.7 | 6.8 | 6.6 |
| #Average words per conversation | 64.1 | 59.1 | 57.5 |
| #Average words per utterance | 8.4 | 8.8 | 8.7 |
| #Average duration of a conversation | 18.5s | 18.4s | 17.8s |
| #Average duration of an utterance | 2.4s | 2.7s | 2.8s |
| #Average turns per conversation | 3.7 | 3.3 | 3.2 |

Table 3: The dataset format. Notations: C_ID = conversation ID, U_ID = utterance ID, Prd. = Pride, Lov. = Love, StartTime and EndTime are in hh:mm:ss, ms format, $\rightarrow$ denotes the directional correlation across two tasks.

| C_ID | U_ID | Utterance | Speaker | StartTime | EndTime | Emo. | Sen. | Sar. | Hum. | Emo.→Sen. | Sar.→Hum. | Prd. | Lov. |
|---|---|---|---|---|---|---|---|---|---|---|---|---|---|
| C_12 | U_4 | Please lend me another two hundred yuan | Wei Zhang | 00:00:07,900 | 00:00:09,480 | Neu. | Neu. | None | Non. | 2 | 0 | Non. | Non. |
| C_12 | U_5 | Why don't you buy a piece of tofu and kill yourself on it? | Yumo Qin | 00:00:10,050 | 00:00:12,854 | Ang. | Neg. | Sar. | Hum. | 2 | 2 | Non. | Non. |

To fulfill the requirements, we choose eighteen famous Chinese TV series as our domain. They consist of multi-speaker conversations with utterances in the forms of text (t), video (v) and audio (a), and cover different genres ($viz.$ comedy, metropolitan opera, drama, crime thriller) and styles ($viz.$ costume, war, idol, history, romance, family, crime). Furthermore, the conversations are extracted from all the episodes of different seasons and therefore cover various topics from daily events to political conflicts (c.f. Tab.1 in supplementary document).

In order to filter out noisy and irrelevant samples, we partition episodes into *short* ($[2s, 7s]$), *medium* ($[8s, 13s]$), *long* ($[14s, 19s]$) and *super long* ($[20s, 25s]$) conversations based on the time intervals of video segments, and randomly sample raw videos from each group. As a result, over 3,800 multi-modal videos are gathered. They are then filtered by the following rules: (1) the video should not contain low-resolution or blank frames; (2) speakers speak clearly in standard mandarin; (3) The speaker's face and voice must appear simultaneously and remain for a certain period of time; (4) The whole conversation in the video should take place in a single scene; and (5) There should be no ambiguity in human annotation (See Section 2.3).

After this step, we obtain a total number of 3,000 video conversations. We extract all the subtitles and transcripts for each conversation with their respective timestamps through Google cloud transcription service[3], and utilize Adobe Premiere Pro[4] to crop the conversation at the utterance level according to the starting and ending timestamps of each utterance. In line with previous studies [9, 13], the heuristic constraints are proposed to accomplish the conversation segmentation: *(1) the timestamps of the utterances in a conversation should be in ascending order. (2) The utterances in a conversation should be from the same episode and scene only.*

After segmentation, we obtain utterance-level aligned text, audio and visual clips of each conversation. The CMMA dataset contains 3,000 multi-party conversations, 21,795 multi-modal utterances and 185,442 word occurrences (license terms please refer to App. 1.3 in Supp.). A conversation has an average number of 7.0 utterances and 60.2 words. Each utterance lasts an average of 2.6 seconds. More statistics can be found in Table 2.

**Data Format.** Each utterance is uniquely identified by a conversation ID and an utterance ID. Its text, video and audio clips are stored in .CSV, .MP4, .WAV files. For examples, see Table 3.

---

[3] https://cloud.google.com/speech-to-text
[4] https://www.adobe.com/products/premiere.html

## 2.2 Label Selection and Annotation

**Multi-affect annotations.** We annotate each utterance with sentiment, emotion, sarcasm and humor labels. For sentiment, apart from the common 3-level annotations (*positive*, *neutral*, *negative*), we present two novel social needs-oriented fine-grained sentiments, *pride* and *romantic love*. In psychology and philosophy, pride is defined as a complex secondary sentiment that involves a feeling of deep pleasure or satisfaction derived from one's own importance [14]. We consider *pride* and *non-pride* as the labels. The triangular theory of love [15, 16] considers romantic love (love for short) as an intense feeling of deep attraction towards another person. It identifies four forms of love, i.e., *immediate love*, *growing love*, *empty love* and *non-love*. We thus take them as love labels[5]. For emotion, Ekman's six universal emotions [13], $viz.$ *joy*, *sadness*, *anger*, *disgust*, *fear*, *surprise*, plus the *neutral* emotion are selected. We perform 2-level annotation for for both sarcasm and humor, i.e., *sarcastic* versus *non-sarcastic* and *humor* versus *non-humor*.

**Cross-task Relevance.** We make the first attempt to annotate the relevance between different affects. Emotion and sentiment are direct expression forms of affect while sarcasm and humor are below-the-surface affect types. The challenges for identifying the inter-relatedness between affects mainly reside in the relevance judgment between pairs of affects of the same level, i.e., emotion $vs$ sentiment, and sarcasm $vs$ humor. Therefore, we perform human annotations on the relevance judgment of the two affect pairs. Essentially, a 5-level annotation in $[-2, -1, 0, 1, 2]$ is used, where the sign stands for whether an affect contributes to the other or the other way around[6], and the absolute value indicates the strength of contribution. For instance, 2 means emotion plays a significant role in sentiment analysis. 0 means the two tasks are irrelevant. Such information can help determine the main and auxiliary tasks, which instructs the design of the multi-task model structure. Its empirical benefits are validated in Section 4.2.

## 2.3 Human Annotation Procedure

We developed a Java-based interface for human annotation. It contains an utterance's video and its speaker and transcript (c.f. Fig.2 in App.A in Supp.). Single-choice questions of affect judgments are presented to annotators. An annotation can click "click to view context" to start watching the video of the target utterance. The interface also contains buttons to submit the annotations, move to the next utterance, or clear all annotations on the page.

The annotation procedure consists of two phases: annotation and re-annotation. Specially, we recruit nine well-educated volunteers including seven undergraduate and two master's students to take part in data annotation and re-annotation. They all signed on the consent form before the study and were paid an equal $7.5/hour in local currency. Prior to annotation, they received professional guidance[7] covering the use of the annotation system, the criteria for labeling, human affect-related knowledge, positive and negative examples, etc. We answered further questions from the volunteers regarding the guidance. Then, they were instructed to annotate 100 examples first to strengthen the inter-annotator agreement, which should reach 90% in principle.

Seven annotators were randomly assigned with annotation tasks. Since human affects are usually intertwined, each annotator was asked to perform the complete set of annotations, including six affect judgment tasks and two relevance judgment tasks described in Section 2.2. The gold standard labels of each utterance are determined by majority voting on all human annotations.

The re-annotation phase started when there was a strong disagreement among seven annotators (e.g., the voting result is 3:3:1 or 2:2:2:1). Two new annotators were asked to re-annotate the disputed samples. We added their voting results to the existing annotations to decide the final golden label. If their votes disagreed, we removed the conversation containing the utterance from the dataset.

## 2.4 Quality Control and Inner Agreement

To guarantee high-quality annotations, we calculate the average agreement rate among nine annotators via the percent agreement calculation approach [17]. The average agreement rates for eight tasks are 88.8%, 71.5%, 86.8%, 94.5%, 82.5%, 94.9%, 69.7% and 67.4%. To increase the

---

[5]The explanation of motivation of pride and love annotation is provided in App.K in supplementary document.

[6]$emotion \rightarrow sentiment$ and $sarcasm \rightarrow humor$ are the main directions. For $A \rightarrow B$, a positive value means A contributes to B.

[7]The guidance and the informed consent are available on https://github.com/annoymity2022/Chinese-Dataset.

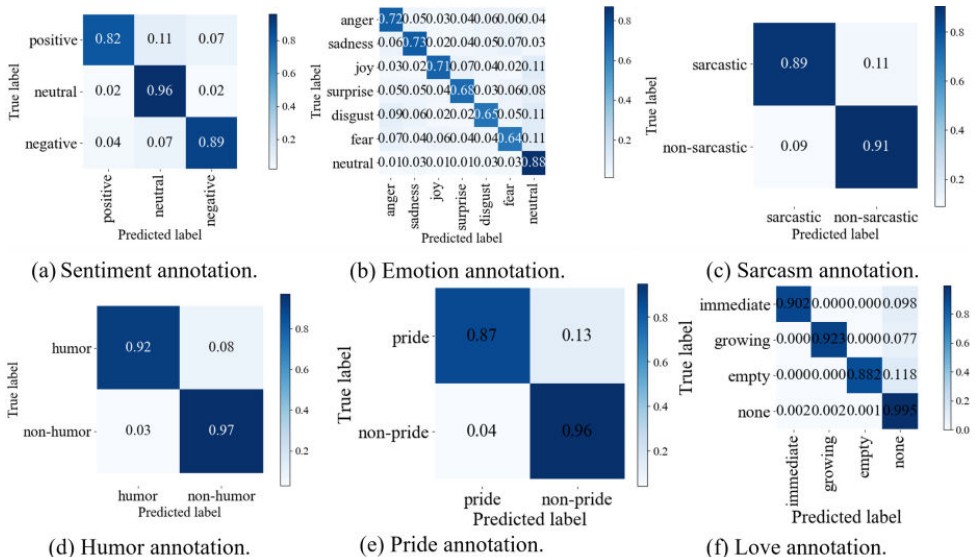

(a) Sentiment annotation.  (b) Emotion annotation.  (c) Sarcasm annotation.

(d) Humor annotation.  (e) Pride annotation.  (f) Love annotation.

Figure 2: The confusion matrices show the annotations difference between different labels for six affects.

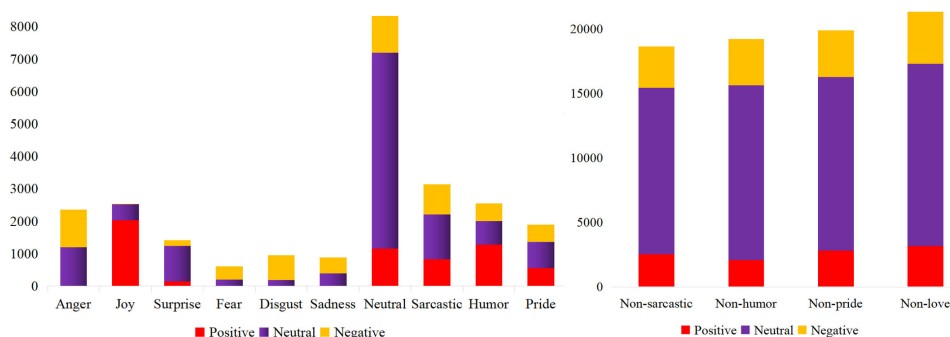

Figure 3: Sentimental distribution of emotion, sarcasm, humor, pride and love samples.

credibility, we computed the Fleiss' kappa score [18]. The overall agreement scores of the annotation are $\kappa = 0.85$, $\kappa = 0.69$, $\kappa = 0.68$, $\kappa = 0.85$, $\kappa = 0.71$, $\kappa = 0.83$, $\kappa = 0.62$, $\kappa = 0.60$ respectively, which means the nine participators have reached substantial agreement on eight annotation tasks. Compared with other related datasets, we have attained the highest inter-agreement scores on all tasks (e.g., the sentiment kappa of MOSI is 0.77, the emotion kappa of MELD is 0.43, the sarcasm kappa of MUStARD is 0.58). We have recruited the highest number of annotators to enhance the quality[8].

Moreover, the confusion matrices are calculated over seven annotators' annotations to present the differences between different affect labels. From Figure 2 (a), one could easily distinguish positive from negative sentiment, but it is relatively difficult to tell neutral sentiment apart from either of them. Similarly, Figure 2 (b) supports the above argument with a small distance between neutral and other emotions, especially sadness, joy and surprise. Figure 2 (c) suggests a minor confusion between sarcastic and non-sarcastic labels. From Figure 2 (d), (e) and (f), we see that the annotators are able to correctly identify humor, pride and love.

## 3 Analysis of CMMA Dataset

**Sentiment distribution over other affects.** Figure 3 presents the distribution of *positive*, *negative* and *neural* samples in other affect labels. (a) For emotion samples, negative sentiment takes up

---

[8]App.B in Supp. provides a systematic comparison between CMMA and other datasets.

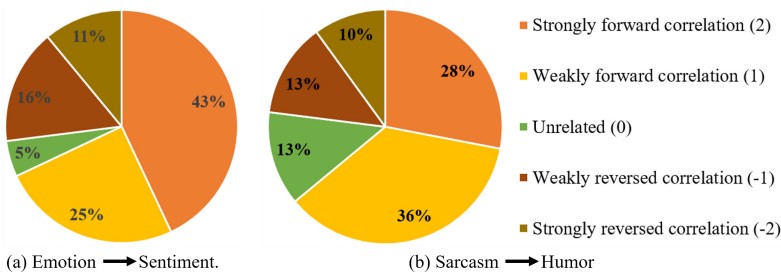

Figure 4: The relevance intensity across tasks.

the majority of anger, fear, disgust and sadness samples, at a ratio of 50.4%, 67.5%, 79.7% and 55.3%, respectively. 80.0% joy samples are of a positive sentiment, confirming with our intuition that joyful expression implies positive sentiment. Since surprise can be expressed with a positive or negative sentiment, positive and negative sentiments have close proportions in surprise samples. Quite surprisingly, about 17.8% samples with neutral emotion have a positive and negative sentiment, which indicates neutral emotion does not necessarily mean neutral sentiment. The results have verified our previous arguments that sentiment and emotion are distinct (See Section 1). (b) Positive (29.9%) and negative (26.4%) sentiments take close proportions of sarcastic samples, but still more samples (43.7%) are of neutral sentiment. (c) Out of humor samples, positive sentiment has the largest proportion (50.4%) than the other two sentiments. This shows that humor tends to provoke positive feelings. (d) Pride samples consist of almost equal numbers of positive, neutral and negative sentiments. One possible reason is that pride is a complicated affect type, and can appear under positive, negative or neutral sentiment. The distribution of love samples is not plotted due to too few samples in the dataset. To sum up, a close relationship between sentiment and other affects is shown in the distributions.

**Affect inter-relatedness.** Figure 4 illustrates the relevance intensity between emotion and sentiment, and sarcasm and humor. We observe that emotion and sentiment are considered as related in 95% samples. The annotators argue that the result of emotion annotation offers great or less help to sentiment judgment in 68% utterances. In the remaining 27% samples, the result of sentiment judgment will help emotion annotation. Similarly, sarcasm and humor are annotated as correlated in 87% samples. The annotators argue that sarcasm judgment tends to offers greater help to humor annotation than humor detection to sarcasm understanding (64% $vs$ 23%). Such annotation helps determine the main and auxiliary tasks in the multi-task learning paradigm (See Section 4.1).

**Topic distribution.** We generate a word cloud via Jieba[9], to visually present 50+ topics of multi-modal conversations in CMMA dataset, as shown in Figure 5 in App.E. We observe that the conversations cover a broad range of ordinary topics, such as daily life, party, work, family, war, idle chat, love, as well as a few relatively specific topics, such as salary and kidnapping. The coverage and distribution of topics illustrate that the CMMA dataset meets the rule of "representativeness".

**Speaker distribution.** CMMA dataset contains around 500 speakers in total. Therefore, we focus solely on leading characters and plot the distributions of different affect types for each of them in Figure 3 (c.f. App. E.2 in Supp.). We observe that positive, joy, sarcastic and humor affects are more likely to happen in the characters of sitcoms. These characters produce plenty of profession-related punchlines. Anger and fear emotions occur more frequently on the tragic figures of dramas and crime thrillers. This implicates that the speaker's information is also valuable for affect judgment. The overall topic and speaker distribution and analysis are detailed in App.E in supplementary document.

# 4 Experiments and Evaluation

## 4.1 Experiment Settings

**Dataset Split.** We randomly split the utterances of CMMA dataset into train, validation and test subsets by 60%, 20%, 20%. The detailed statistics in App.G.

---

[9]https://github.com/fxsjy/jieba

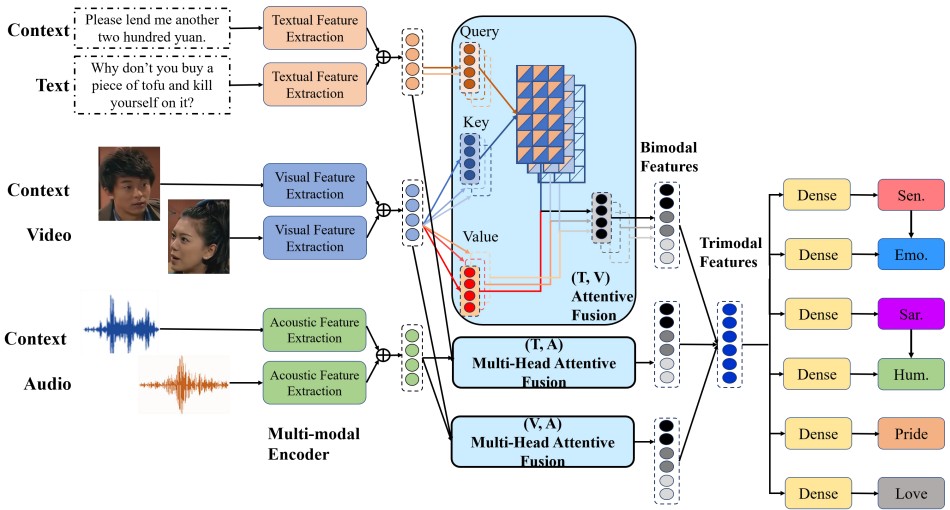

Figure 5: Multi-modal multi-affect joint detection model.

**Model architecture.** We build a model that seeks to classify all the six affect types, i.e., sentiment, emotion, sarcasm, humor, pride and love, with different combinations of input features. Figure 5 presents the multi-task model architecture. The text, video and audio inputs are each passed to an encoders to extract their features. We consider four neural text encoders, i.e., bidirectional LSTM (BiLSTM), BERT [19] and GPT-2 [20] or GPT-3[10] [21]. For video, two widely used visual encoders, i.e., EfficientNet [22] and ResNet are selected. For audio encoder, the pre-trained VGGish network are used. For each modality, the encoded utterance is concatenated with its encoded context, and the unimodal contextual features are combined by multi-modal fusion. Specific multi-modal fusion strategies include multi-head attentive fusion, concatenation, addition, element-wise multiplication and maximum. The obtained multi-modal representation is then passed through task-specific dense layers for each affect detection task. The labels of all tasks are produced in the forward pass, where we set different weights for different tasks.

On top of the multi-task structure, we add connections between dense layers of different tasks in the form of weighted sum, base on the overall statistics of the relevance intensity value. For example, if the $emotion \rightarrow sentiment$ relevance intensity is -2, the sentiment dense layer will be directly added to the emotion dense layer. If the $emotion \rightarrow sentiment$ relevance intensity is -1, the sentiment dense layer will be halved before adding with the emotion dense layer. We adopt *precision* (P), *recall* (R), *macro-F1* ($M_a$-F1) and *balanced accuracy* (acc) as the evaluation metrics.

## 4.2  Results and Discussion

We present the experimental results in Table 4. For text encoder, BiLSTM underperforms BERT and GPT on all tasks but pride detection. Between the contextualized encoders, GPT-3 outperforms BERT for sarcasm detection task, and GPT-2 and GPT-3 slightly outperform BERT for love classification task, but are weaker than BERT for other four tasks. Therefore, BERT performs better for most classification tasks. For video encoder, ResNet beats EfficientNet by a large margin. Among the unimodal classifiers, the pre-trained VGGish acoustic encoder performs significantly worse than the best-performed encoders for the other modalities, possibly due to the difficulty of extracting acoustic features. For the same reason, Text+Video is the best-performed bi-modal setup, where BERT+ResNet occupies the best performance on almost all tasks. In addition, Text+Audio has an overall better performance than Video+Audio. Finally, the trimodal setups significantly outperform bimodal setups, where BERT+ResNet+VGGish performs the best.

The performance gaps between trimodal and bimodal features illustrate the complementary nature of all three modalities. Still, the BERT-encoded textual feature plays a major contribution to the overall performance. We posit that the reasons are two fold. First, text tends to directly influence affect

---

[10]Since GPT-3 is not open source, we only use it to perform text sentiment analysis, instead of constructing the multi-modal fusion model.

Table 4: Comparison of different models.

| Model | Text | Video | Audio | Sentiment | | | Emotion | | | Sarcasm | | | Humor | | |
|---|---|---|---|---|---|---|---|---|---|---|---|---|---|---|---|
| | | | | P | R | $M_a$-F1 | P | R | $M_a$-F1 | P | R | $M_a$-F1 | P | R | $M_a$-F1 |
| Text | BiLSTM | - | - | 50.36 | 51.22 | 50.74 | 41.52 | 62.12 | 44.74 | 56.43 | 50.64 | 52.29 | 43.69 | 55.9 | 49.05 |
| | BERT | - | - | 56.77 | 55.51 | 54.89 | 51.85 | 70.87 | 56.18 | 54.56 | 53.88 | 53.61 | 51.5 | 56.94 | 54.08 |
| | GPT-2 | - | - | 53.88 | 58.01 | 54.35 | 45.33 | 44.37 | 45.21 | 51.41 | 53.48 | 52.42 | 44.81 | 64.39 | 52.85 |
| | GPT-3 | - | - | 54.66 | 54.21 | 54.43 | 48.72 | 47.65 | 48.18 | 53.27 | 54.87 | 54.06 | 49.84 | 47.21 | 48.49 |
| Video | - | EfficientNet | - | 42.86 | 45.12 | 42.84 | 38.08 | 61.58 | 42.18 | 46.77 | 61.66 | 53.19 | 38.06 | 52.8 | 44.23 |
| | - | ResNet | - | 48.92 | 51.53 | 49.40 | 47.65 | 47.89 | 47.66 | 57.66 | 57.84 | 57.75 | 41.84 | 55.69 | 47.78 |
| Audio | - | - | VGGish | 41.15 | 62.12 | 44.89 | 33.24 | 26.70 | 30.64 | 42.19 | 43.54 | 42.85 | 34.98 | 44.81 | 46.84 |
| Text+Video | BiLSTM | EfficientNet | - | 49.68 | 52.33 | 50.20 | 40.51 | 39.69 | 40.10 | 45.70 | 57.17 | 50.80 | 44.87 | 58.18 | 50.53 |
| | BiLSTM | ResNet | - | 48.77 | 51.27 | 49.30 | 36.68 | 48.86 | 37.49 | 50.69 | 57.17 | 53.74 | 42.51 | 61.7 | 50.34 |
| | BERT | EfficientNet | - | 65.47 | 69.88 | 66.75 | 41.16 | 61.68 | 44.29 | 55.74 | 58.35 | 57.02 | **53.59** | 61.9 | **58.44** |
| | BERT | ResNet | - | **67.32** | **73.36** | **68.89** | **56.24** | 68.54 | **57.82** | **67.84** | 65.69 | **66.75** | 52.03 | 66.25 | 58.29 |
| | GPT-2 | EfficientNet | - | 58.13 | 64.24 | 59.17 | 38.08 | 61.58 | 42.18 | 45.45 | 56.05 | 50.20 | 46.02 | 63.35 | 53.31 |
| | GPT-2 | ResNet | - | 59.09 | 66.32 | 60.03 | 42.17 | 61.80 | 45.91 | 50.55 | 61.65 | 55.56 | 45.75 | 64.6 | 53.56 |
| Video+Audio | - | EfficientNet | VGGish | 49.22 | 50.21 | 48.27 | 41.15 | 62.12 | 44.89 | 38.59 | 59.19 | 46.73 | 40.89 | 64.18 | 49.96 |
| | - | ResNet | VGGish | 52.47 | 53.52 | 51.62 | 52.12 | 51.04 | 51.44 | 42.12 | 58.74 | 49.06 | 42.63 | 65.84 | 51.75 |
| Text+Audio | BiLSTM | - | VGGish | 46.97 | 49.55 | 46.84 | 43.13 | 64.83 | 46.82 | 40.85 | 66.59 | 50.64 | 42.23 | 64.18 | 50.94 |
| | BERT | - | VGGish | 54.41 | 55.25 | 55.74 | 46.93 | 63.31 | 50.36 | 43.57 | 68.39 | 53.22 | 48.99 | 65.22 | 55.95 |
| | GPT-2 | - | VGGish | 51.41 | 53.48 | 52.42 | 45.23 | 66.98 | 49.44 | 41.52 | **69.73** | 52.05 | 45.29 | 63.77 | 52.97 |
| Text+Video+Audio | BERT | EfficientNet | VGGish | 69.59 | 73.98 | 71.12 | 53.03 | 74.37 | 57.36 | 69.38 | 65.02 | 67.13 | 63.76 | 69.57 | 66.53 |
| | BERT | ResNet | VGGish | 71.64 | **76.31** | **73.29** | **56.71** | **76.32** | **61.76** | **76.28** | **74.22** | **75.23** | **76.47** | **75.36** | **75.91** |
| | GPT-2 | EfficientNet | VGGish | 65.66 | 69.47 | 66.86 | 47.06 | 73.71 | 51.49 | 58.95 | 62.78 | 60.8 | 58.16 | 62.73 | 60.36 |
| | GPT-2 | ResNet | VGGish | **71.76** | 74.87 | 72.88 | 52.09 | 73.82 | 56.17 | 74.44 | 67.26 | 70.67 | 65.80 | 73.29 | 69.34 |
| Trimodal *vs* Bimodal (%) | - | - | - | +6.6 | +4.0 | +6.4 | +0.8 | +11.3 | +6.8 | +12.4 | +6.4 | +12.6 | +42.6 | +13.7 | +29.6 |

Table 5: Comparison of different multi-modal fusion strategies.

| Trimodal Accuracy | Sentiment | | Emotion | | Sarcasm | | Humor | |
|---|---|---|---|---|---|---|---|---|
| | Validation | Test | Validation | Test | Validation | Test | Validation | Test |
| Multi-head Attention | 74.81 | **78.48** | 72.24 | **77.09** | **82.44** | **85.64** | 84.31 | **86.15** |
| Concatenate | **76.76** | 76.31 | 73.14 | 76.32 | 82.22 | 84.28 | **85.06** | 85.88 |
| Add | 71.62 | 77.39 | 73.33 | 76.36 | 82.37 | 84.86 | 85.06 | 82.93 |
| Multiply | 69.85 | 72.22 | 70.39 | 73.05 | 78.77 | 78.54 | 80.91 | 81.31 |
| Maximum | 75.95 | 76.38 | **74.11** | 72.47 | 81.25 | 83.13 | 81.66 | 79.42 |

understanding, while visual and acoustic signals are on a higher level of abstraction. Second, BERT can effectively capture the word dependencies and extract contextualized features. The comparison of different fusion strategies are given in Table 5, where multi-head attention performs best for all tasks. The reason is that the attention mechanism is a best fit in that it automatically learns to pay attentions to different modalities for different utterances.

**Effect of Conversational Context.** We examined the effect of conversational context by choosing different context lengths of $[0, 1, 2]$ and construct speaker-aware and topic-aware models on top of the main model architecture. From Table 6, we observe that the model with two contexts yield the best performance, and both speaker-aware and topic-aware settings beat the vanilla settings. All the results demonstrate a crucial role of conversational context, which motivates improved conversational context modeling strategies for future work. By simply merging two contexts with the target utterance, the strong model has obtained significant improvement against the previous setup with zero context. We argue that our setup provides a strong baseline for multi-affect detection, and it may achieve better classification performance if superior attempts on context modeling have been done.

**Effect of Cross-task Relevance.** We investigate the impact of the relevance between sentiment-emotion, sarcasm-humor for affect detection by jointly detecting the two affects in each pair. We compare our setup with single-task learning (STL) framework and the standard multi-task learning (S-MTL) paradigm for jointly addressing both tasks. For our relevance-aware models (RaM), the emotion dense layer is added to sentiment layer, while the sarcasm dense layer is halved and added to the humor dense layer, based on the statistics in Figure 4. We show their classification performance in Table 7. We notice that both multi-task models outperform single-task learning framework. Furthermore, our model consistently outperforms the S-MTL setting, which indicates that the intensity value effectively captures the cross-task relevance and brings benefit to the multi-task learning paradigm. The success of the simple strategy demonstrates the enormous potential of cross-task relevance annotations. We expect to see greater performance gains to multi-affective joint detection by improved leverage of the relevance annotations on a finer-grained level.

**Linguistic Insights.** We make detailed linguistic analysis in App. L.

Table 6:  Effect of context, speakers and topics.

| Model | Range | Sentiment | | Emotion | | Sarcasm | | Humor | |
|---|---|---|---|---|---|---|---|---|---|
| | | $M_a$-F1 | Acc | $M_a$-F1 | Acc | $M_a$-F1 | Acc | $M_a$-F1 | Acc |
| Context | 0 | 73.29 | 76.31 | 61.76 | 76.32 | 75.23 | 85.64 | 75.91 | 86.15 |
| | 1 | 71.36 | 73.58 | 65.81 | 77.97 | 71.97 | 84.99 | 71.01 | 84.94 |
| | 2 | **76.91** | **78.12** | **68.79** | **79.02** | **75.72** | **87.76** | **76.06** | **87.97** |
| Speaker | No Speaker | 73.29 | **76.31** | 61.76 | 76.32 | 75.23 | 85.64 | **75.91** | **86.15** |
| | Speaker Aware | **74.39** | 75.24 | **64.23** | **76.39** | **75.27** | **87.33** | 72.06 | 85.22 |
| Topic | No Topic | 73.29 | 76.31 | 61.76 | 76.32 | **75.23** | **85.64** | 75.91 | 86.15 |
| | Topic Aware | **76.62** | **76.71** | **63.31** | **78.14** | 72.63 | 85.37 | **76.02** | **87.63** |

Table 7: Effect of the relevance between sentiment-emotion / sarcasm-humor.

| Setup | Sentiment | | Emotion | | Sarcasm | | Humor | |
|---|---|---|---|---|---|---|---|---|
| | $M_a$-F1 | Acc | $M_a$-F1 | Acc | $M_a$-F1 | Acc | $M_a$-F1 | Acc |
| STL | 71.17 | 72.22 | 59.75 | 72.47 | 71.97 | 83.13 | 73.21 | 79.41 |
| $S - MTL : Emo.$ | 71.61 | 72.85 | **61.76** | 76.32 | - | - | - | - |
| $S - MTL : Sen.$ | 73.14 | 75.52 | 60.12 | 73.39 | - | - | - | - |
| **RaM** | **74.31** | **79.55** | **61.76** | **77.09** | - | - | - | - |
| $S - MTL : Sar.$ | - | - | - | - | 74.22 | **85.64** | 73.77 | 80.46 |
| $S - MTL : Hum.$ | - | - | - | - | 72.27 | 83.84 | 74.51 | 85.42 |
| **RaM** | - | - | - | - | **75.23** | **85.64** | **75.91** | **86.15** |

# 5   Conclusions and Future work

Few works (including the recent large language models) have set foot in multi-affect joint detection in conversations, largely due to the lack of multi-modal conversation datasets with multi-affect annotations. We have filled this gap by proposing CMMA, the first multi-modal multi-affect conversation dataset. CMMA consists of 21,795 multi-modal utterances from 3,000 multi-party conversations. Apart from rich affect labels including sentiment, emotion, sarcasm and humor, the dataset contains annotation relevance between affect types. We have performed comprehensive qualitative and quantitative studies for analyzing the dataset, and presented a range of baselines to evaluate the potential of CMMA. The results demonstrate the quality of the dataset and indicate the need of novel investigations in models in multi-modal multi-affect joint detection in conversations.

**Limitations and Potential Risks.** (1) The created dataset does not reach the requirement of "balance". The CMMA dataset also has its unique characteristics and limitations that may affect the generalization of our results. The dataset's focus on Chinese conversations from TV shows may limit the direct application of our findings to other languages. We encourage future researchers to consider cross-linguistic validation. (2) The created dataset does not reach the requirement of "balance". The implicit bias (e.g., age-related bias, profession-related bias) may be introduced. In addition, the potential biases that may arise when sourcing content from TV shows, where the emotion expressed by the actors may be exaggerated. We will attempt to design a bias mitigating approach to check and alleviate the impact of such biases in the future work. The researchers should understand these limitations and take them into careful consideration.

## Acknowledgements

This work is partly supported by a Project of Strategic Importance grant of The Hong Kong Polytechnic University (project no. 1-ZE2Q). This paper is also partly supported by a grant under Hong Kong RGC Theme-based Research Scheme (project no. T45-401/22-N). This work is also supported by National Science Foundation of China under grant No. 62006212, Fellowship from the China Postdoctoral Science Foundation (2023M733907). This work is funded in part by Beijing Municipal Natural Science Foundation (grant no: 4222036).

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
