# CMMA : Benchmarking Multi-Affection Detection in Chinese Multi-Modal Conversations

# Supplementary Document

# 1 Datasheets for datasets

## 1.1 Motivation

The recent popularity of the multi-task learning paradigm led to an emerging focus on multi-affection joint detection. However, the datasets they worked on, such as MELD [1], IEMOCAP [2], UR-FUNNY [3], MUStARD [4], etc., have annotations on solely one or two types of affection, and inter-relatedness between tasks is absent. Without an explicit annotation of cross-task correlations, the potential of multi-modal multi-affection joint detection could not be fully explored, neither deepen the understanding on human complicated affections.

We fill the gap by constructing a large-scale benchmark multi-modal multi-affection conversational dataset. We manage to tackle the following main challenges for building such a dataset: (1) *multi-affection joint judgment*: the subjectivity and creativity of human language make it hard to judge different affections at the same time accurately; (2) *multi-affection correlation*: different affections can be indistinguishable at certain circumstances, and it is difficult to accurately measure their relatedness; (3) *context effect*: an utterance may express different affections in different conversational contexts.

## 1.2 Composition

CMMA consists of 21,795 multi-modal utterances from 3,000 multi-party conversations. Each utterance is annotated with sentiment (including pride and romantic love), emotion, sarcasm and humor labels, accompanied by sentiment-emotion and sarcasm-humor inter-relatedness measures. Considering that the external knowledge implicitly influences the speaker's affective state, the speaker's background (i.e., name, profession, sex, personality) and the topic of each conversation are provided, an example as illustrated in Figure 1. Each utterance contains textual, visual and acoustic information, which are stored in .CSV, .MP4, .WAV files.

We randomly split the utterances of CMMA dataset into train, validation and test subsets by 60%, 20%, 20%. Table 4 show the statistics.

## 1.3 Collection Process and License

In this work, eighteen famous Chinese TV series, e.g., "爱情公寓" (iPartment), "三国演义" (Romance of Three Kingdoms), "武林外传" (My Own Swordsman), etc., are chosen as our domain[1]. Such TV series consist of multi-speaker conversations with utterances in forms of text (t), video (v) and audio (a), and cover different genres (*viz.* comedy, metropolitan opera, drama, crime thriller) and

---

[1]We have also collected real dialogue samples from the 10086 customer service of China Mobile Communication Group Tianjin Co., but due to the protection of user privacy and company regulations, we cannot publicly disclose these samples.

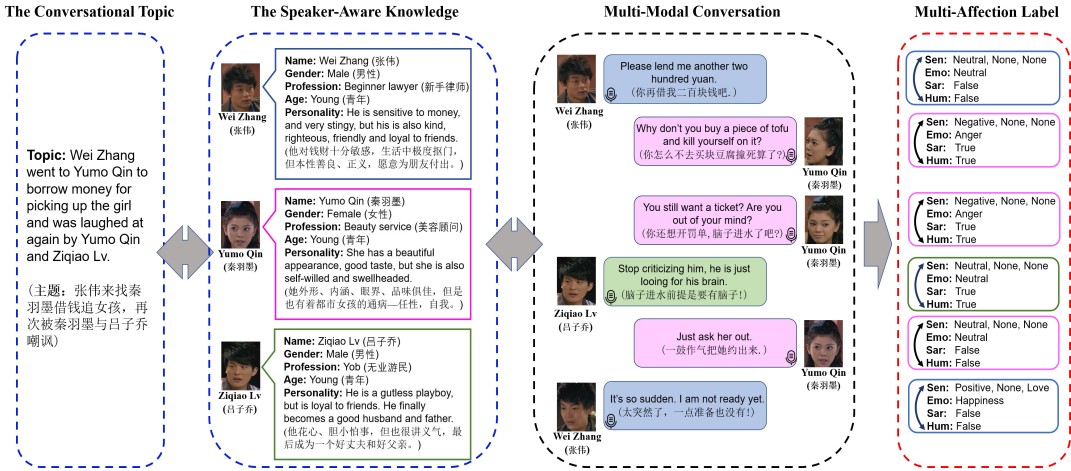

**The Conversational Topic**

**Topic:** Wei Zhang went to Yumo Qin to borrow money for picking up the girl and was laughed at again by Yumo Qin and Ziqiao Lv.

（主题：张伟来找秦羽墨借钱追女孩，再次被秦羽墨与吕子乔嘲讽）

**The Speaker-Aware Knowledge**

Wei Zhang (张伟)
**Name:** Wei Zhang (张伟)
**Gender:** Male (男性)
**Profession:** Beginner lawyer (新手律师)
**Age:** Young (青年)
**Personality:** He is sensitive to money, and very stingy, but his is also kind, righteous, friendly and loyal to friends.
（他对钱财十分敏感，生活中极度抠门，但本性善良、正义，愿意为朋友付出。）

Yumo Qin (秦羽墨)
**Name:** Yumo Qin (秦羽墨)
**Gender:** Female (女性)
**Profession:** Beauty service (美容顾问)
**Age:** Young (青年)
**Personality:** She has a beautiful appearance, good taste, but she is also self-willed and swellheaded.
（她外形、内涵、眼界、品味俱佳，但是也有着都市女孩的通病—任性，自我。）

Ziqiao Lv (吕子乔)
**Name:** Ziqiao Lv (吕子乔)
**Gender:** Male (男性)
**Profession:** Yob (无业游民)
**Age:** Young (青年)
**Personality:** He is a gutless playboy, but is loyal to friends. He finally becomes a good husband and father.
（他花心、胆小怕事，但很讲义气，最后成为一个好丈夫和好父亲。）

**Multi-Modal Conversation**

Wei Zhang (张伟): Please lend me another two hundred yuan. （你再借我二百块钱吧。）

Yumo Qin (秦羽墨): Why don't you buy a piece of tofu and kill yourself on it? （你怎么不去买块豆腐撞死算了？）

Yumo Qin (秦羽墨): You still want a ticket? Are you out of your mind? （你还想开罚单，脑子进水了吧?）

Ziqiao Lv (吕子乔): Stop criticizing him, he is just looing for his brain. （脑子进水前提是要有脑子！）

Yumo Qin (秦羽墨): Just ask her out. （一鼓作气把她约出来。）

Wei Zhang (张伟): It's so sudden. I am not ready yet. （太突然了，一点准备也没有!）

**Multi-Affection Label**

**Sen:** Neutral, None, None
**Emo:** Neutral
**Sar:** False
**Hum:** False

**Sen:** Negative, None, None
**Emo:** Anger
**Sar:** True
**Hum:** True

**Sen:** Negative, None, None
**Emo:** Anger
**Sar:** True
**Hum:** True

**Sen:** Neutral, None, None
**Emo:** Neutral
**Sar:** True
**Hum:** True

**Sen:** Neutral, None, None
**Emo:** Neutral
**Sar:** False
**Hum:** False

**Sen:** Positive, None, Love
**Emo:** Happiness
**Sar:** False
**Hum:** False

Figure 1: An example of multi-modal conversation from the CMMA dataset and each utterance is annotated with sentiment, emotion, sarcasm, humor, pride and love labels.

Table 1: Statistics of CMMA Dataset.

| TV Show | Genre | Style |
|---|---|---|
| "武林外传" (My Own Swordsman) | Comedy | Costume |
| "爱情公寓" (iPartment) | Comedy | Idol, Romance |
| "地下交通站" (The Safe House) | Comedy | War |
| "炊事班的故事" (The Story of Cooking Class) | Comedy | War |
| "家有儿女" (Home with Kids) | Comedy, Metropolitan opera | Family |
| "媳妇的美好时代" (Beautiful Daughter-in-Law) | Metropolitan opera | Romance, Family |
| "欢乐颂" (Ode to Joy) | Metropolitan opera | Romance, Idol |
| "都挺好" (All Is Well) | Metropolitan opera | Family |
| "三国演义" (Romance of Three Kingdoms) | Drama | War, History |
| "父母爱情" (Romance of Our Parents) | Drama | Romance, Family |
| "人民的名义" (In the Name of People) | Drama | Crime |
| "福贵" (Fu Gui) | Drama | Family, Romance |
| "我的团长我的团" (My Chief and My Regiment) | Drama | War |
| "铁齿铜牙纪晓岚" (Ji Xiaolan) | Comedy | History, Idol |
| "白夜追凶" (Day and Night) | Crime thriller | Crime |
| "心理罪" (Guilty of Mind) | Crime thriller | Crime |
| "天道" (Destiny) | Crime thriller | Crime |
| "隐秘的角落" (The Bad Kids) | Crime thriller | Crime |

styles (*viz.* costume, war, idol, history, romance, family, crime). The statistics of the TV are shown in Table 1. We could notice that our domain includes six comedy shows, four metropolitan opera shows, five dramas and four thrillers, which is well-proportioned (6:4:5:4). Moreover, such TV shows cover various styles, e.g., costume, idol, romance, war, family, history, crime, which provide numerous conversation topics. Both actions will ensure us to collect balanced sentiment, emotion, sarcasm, humor, pride and love labels in the best possible way. We argue that the speaker's information is also collected. But the speakers are virtual characters, where we do not disclose personal information.

***License terms.*** To avoid copyright disputes, we have access to downloadable resources from public platforms. We argue that CMMA is made available for research purposes only. We have designed a proper license (namely CC BY-NC 4.0) attached to the CMMA dataset to clearly describe how to properly and responsibly use the CMMA dataset. It will help guide the user of the CMMA dataset in making informed decisions about how the CMMA dataset can and cannot be used[2].

---

[2]https://github.com/annoymity2022/Chinese-Dataset

## 1.4 Preprocessing/Cleaning/Labeling

**Preprocessing, cleaning and labeling.** See the Section 2 in the main paper.

**The Inter-Agreement Comparison between CMMA and Benchmarking datasets.** Sentiment/emotion annotation is a challenging task, in view of their intrinsic high-level abstraction and subjectivity. In the paper, we have elaborated on our efforts to enhance the reliability of the annotators. For example, (1) the annotators received professional guidance prior to annotation; (2) we made sure all the recruiters' uncertainties about the annotations were properly answered; (3) we provided annotation tips on the platform; (4) the annotators were instructed to annotate 100 examples, etc.

We also compare our inter-agreement (kappa score) with a range of benchmark datasets that reported the inter-agreement. As shown Tab. 2, we have the highest inter-agreement scores on all of eight annotation tasks. We also have recruited the highest number of annotators to enhance the quality. Note that MELD only asked the annotators to annotate the surprise as positive and negative sentiments, where their inter-agreement score will not be adopted to make comparison. UR-FUNNY/MHD/BBT/iSarcasm/Memotion did not report the inter-agreement. Sarcasm Corpus V2 and only reported the percentage agreement (80%), which was lower than ours (86.8%).

Table 2: Inter-agreement comparison between CMMA and other datasets.

| Affection | CMMA (ours) | MOSI | MELD | IEMOCAP | ScenarioSA | MUStARD | EmotionLines | MUMOR | MaSaC | MEISD |
|---|---|---|---|---|---|---|---|---|---|---|
| Sentiment | **0.85** | 0.77 | 0.48 | 0.57 | 0.57 | - | - | 0.84 | - | 0.75 |
| Emotion | **0.69** | - | 0.43 | 0.40 | - | - | 0.33 | 0.45 | - | 0.67 |
| Sarcasm | **0.68** | - | - | - | - | 0.58 | - | - | 0.65 | - |
| Humor | **0.85** | - | - | - | - | - | - | 0.81 | 0.68 | - |
| Pride | **0.71** | - | - | - | - | - | - | - | - | - |
| Love | **0.83** | - | - | - | - | - | - | - | - | - |
| Num. of Annotators | **9** | 5 | 3 | 6 | 5 | 3 | 5 | 3 | 5 | 4 |

## 1.5 Uses and Distribution

We state that the CMMA dataset is suitable for multimodal multi-affective joint detection tasks, limited to sentiment analysis, emotion recognition, sarcasm detection, humor detection, and love and pride recognition tasks in natural language processing, and the copyright of the CMMA data set belongs to the Affective Computing Laboratory of the Software College of Zhengzhou University of Light Industry. Following paper acceptance, the full dataset will be made available at `https://github.com/annoymity2022/Chinese-Dataset`.

## 1.6 Maintenance

As far as the dataset update iteration is concerned, our laboratory will have special personnel to conduct inspection and maintenance every six months, including updating the data set (such as correct labeling errors, add new instances, delete instances). Meanwhile, we welcome other research teams to use this dataset.

## 1.7 Uses

- Has the dataset been used for any tasks already? If so, please provide a description?
  It is proposed use for multi-modal emotion recognition task.

- Is there a repository that links to any or all papers or systems that use the dataset? If so, please provide a link or other access point.
  It is a new dataset. We run existing state-of-the-art models and release the code at Github.com.

- What (other) tasks could the dataset be used for?
  Many other tasks like dialogue understanding and generation can be also used.

- Is there anything about the composition of the dataset or the way it was collected and preprocessed/cleaned/labeled that might impact future uses? For example, is there anything that a future user might need to know to avoid uses that could result in unfair treatment of individuals or groups (e.g., stereotyping, quality of service issues) or other undesirable

Table 3: Dataset nutrition label.

| Name | Description |
|------|-------------|
| C_ID | The ID of the conversation |
| U_ID | The ID of utterance in one conversation |
| Utterance | The textual content of utterance |
| Speaker | Authors of utterances in conversation |
| StartTime-EndTime | Timestamp from start to end of an utterance |
| Emo. | Emotion label (joy, sadness, anger, disgust, fear, surprise or neutral) |
| Sen. | Sentiment label (poitive, neutral or negative) |
| Sar. | Sarcasm label (sarcastic or non-sarcastic) |
| Hum. | Humor label (humor or non-humor) |
| Prd. | Pride label (pride or non-pride) |
| Lov. | Love label (immediate love, growing love, empty love non-love) |
| $Emo. \rightarrow Sen.$ | A positive value means Emotion contributes to Sentiment |
| $Sar. \rightarrow Hum.$ | A positive value means Sarcasm contributes to Humor |
| TopicDescription | The topic elaboration of the conversation |
| Profession | The profession of the speaker |
| Personality | The character of the speaker |

harms (e.g., financial harms, legal risks) If so, please provide a description. Is there anything a future user could do to mitigate these undesirable harms?

N/A

- Are there tasks for which the dataset should not be used? If so, please provide a description. N/A

## 2 Dataset nutrition labels

Table 3 shows different modules of the CMMA dataset nutrition label, together with their description.

## 3 Data Statements for Natural Language Processing

We state the CMMA dataset from the following five aspects:

- Metadata: The full name of the CMMA dataset is A Chinese Multi-Modal Dataset for Multi-Affection Detection in Conversations, mainly used for joint learning of multi-affective tasks such as sentiment, emotion, sarcasm and humor. The copyright belongs to the Affective Computing Laboratory of the Software College of Zhengzhou University of Light Industry.

- The data set is mainly composed of three parts of files (namely .CSV, .MP4, .WAV files), which record text, video and audio information respectively.

- The source of the data set is mainly manual crawling and labeling.

- The quality of the data set through qualitative and quantitative evaluation, please see the Section 2 in the main paper for details.

- The CMMA dataset follows the usage rules and restrictions of the license description dataset, and commercial use, reorganization and conversion are not allowed. If you want to expand, you must get the unity of the copyright owner.

## 4 Data Accessibility

The CMMA dataset is available at `https://github.com/annoymity2022/Chinese-Dataset` and `https://drive.google.com/file/d/19NgqpYLPa3bLFm4YUjuCT2sbmQemziG-/view?usp=sharing`.

# 5    Accountability frameworks

## 5.1    Data collection and processing specifications

We declare that our data collection is all in publicly accessible links and follow the following rules in data processing.

- Data collection and processing should comply with relevant regulations and ethical requirements, and use appropriate technical tools and methods to ensure data quality and integrity.
- Data collection should clarify key information such as data type, source, time, and location, and record the collection personnel and process.
- Data processing should establish a clear data cleaning, conversion and integration process, and carry out data verification and deduplication.
- Data sampling and sample selection should fully consider the impact of research design and sampling error, and carry out statistical inference and reliability analysis.

## 5.2    Dataset usage and evaluation mechanisms

Dataset usage and evaluation mechanisms should follow the following principles:

- The use of data sets should follow relevant laws and regulations, respect data privacy and intellectual property rights, and prevent abuse and discriminatory results.
- Dataset results should be interpreted and applied within a reasonable margin of error, with a full explanation of their limitations and applicability of inferences.
- Dataset users should assign corresponding responsibilities and obligations, including data protection, fair use, and social responsibility.
- The process of using data sets should be regularly audited and evaluated in order to detect and correct problems in a timely manner and improve the value and credibility of datasets.

# 6    Author Statement

We (on behalf of all the authors) statement that we will bear all responsibility in case of violation of rights, etc., and confirmation of the data license.

# 7    Hosting, Licensing, and Maintenance Plan for CMMA Dataset

Our CMMA dataset is a groundbreaking multimodal conversational emotional dataset that we have created and publicly released. To ensure its accessibility and long-term availability, we have developed a comprehensive hosting, licensing, and maintenance plan.

**Hosting:** The CMMA dataset will be hosted on a reliable and secure server infrastructure, i.e., Github.com and Google Drive (see Sec. 4). We will ensure fast and uninterrupted access to the dataset for researchers, developers, and interested parties.

**Licensing:** We have designed a proper license (CC BY-NC 4.0) attached to the CMMA dataset to clearly describe how to properly and responsibly use the CMMA dataset.

**Maintenance:** We are committed to the continuous maintenance and improvement of the CMMA dataset. Regular updates will be provided to address any identified issues, ensure data quality, and incorporate user feedback.

# Appendix

## A. An Example of Multi-Modal Conversation from CMMA

We present an example of multi-modal conversation form the proposed CMMA dataset, as shown in Fig. 1.

The layout of our Java-based annotation interface is illustrated in Fig. 2.

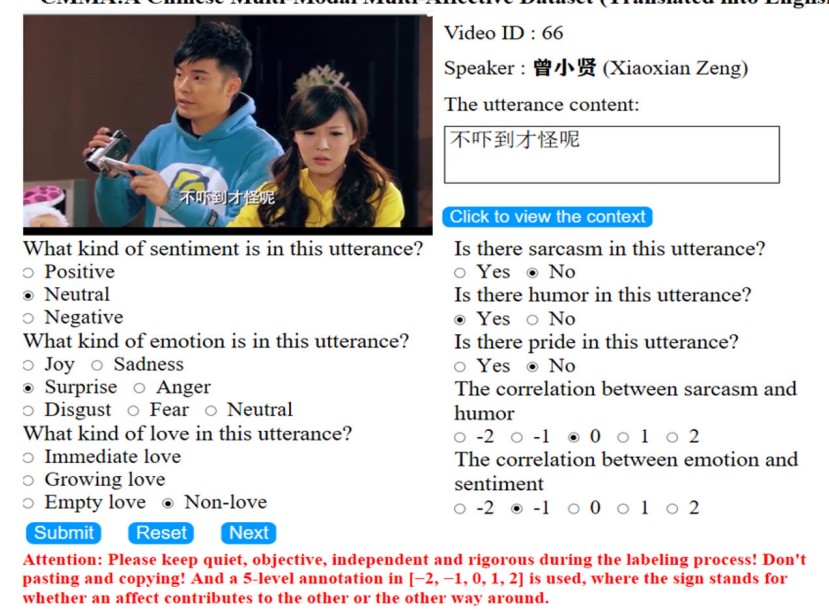

Figure 2: Layout of the annotation interface.

## B. Related Work

### B.1 Multi-Modal Sentiment, Emotion, Sarcasm and Humor Datasets

We present an overview of related resources covering multi-modal sentiment, emotion, sarcasm and humor.

**Sentiment and emotion detection dataset.** Multimedia researchers have created various sentiment and emotion datasets to meet the needs of multi-modal sentiment and emotion analysis, most notably YouTube [5], Getty Image [6], MOUD [7], CMU-MOSI [8] and CMU-MOSEI [9]. These datasets contain human annotations solely on the whole multi-modal data. Yu et al. [10] created CH-SIMS from 2,281 refined Chinese video segments in the wild, where each segment is annotated with both multi-modal and unimodal sentiment labels. With the recent attention to human interaction, a series of conversational sentiment and emotion datasets were presented, such as IEMOCAP [2], MELD [1], MEISD [11], ScenarioSA [12]. IEMOCAP, MELD and MEISD provided both sentiment and emotion annotations, while ScenarioSA only labeled the sentiment polarity. However, they did not pay attention to the inter-relatedness across sentiment and emotion.

**Sarcasm detection dataset.** Although numerous datasets have been proposed for textual sarcasm detection, e.g., iSarcasm [13], SARC [14], there were very few benchmark multi-modal datasets, due to the complexity of multi-modal sarcasm annotation. Three representative multi-modal datasets were proposed, i.e., MUStARD [4], Twitter [15] and Silver-Standard Dataset [16] were typically chosen as test-beds for sarcasm detection models. Moreover, Alnajjar et al. [17] presented the first Spanish multi-modal sarcasm dataset.

**Humor detection dataset.** Humor understanding has gained recent popularity. Notable humor datasets, such as multi-modal Humor Dataset (MHD) [18], Big Bang Theory (BBT) [19], UR-FUNNY [3], were created. Such datasets studied the importance of punchline and context in understanding humor from the perspectives of conversation (sitcoms) and personal presentation (TED videos). But each utterance was only annotated as humor or non-humor.

**Multiple affections detection dataset.** Researcher have created a few multi-affection datasets. Chauhan et al. [20] manually annotated the MUStARD dataset with sentiment and emotion labels. Wu et al. [21] presented MUMOR, a Chinese conversation dataset including humor, sentiment and emotion labels, but the dataset is not publicly accessible. Bedi et al. [22] built MaSaC from English-Hindi TV series, where each utterance is annotated with sarcasm and humor labels. Sharma et al. [23] released the Memotion dataset, including 10K Internet memes labelled with sentiment, emotion, sarcasm and humor. However, it did not involve conversational context or inter-dependency across tasks.

**Differences from existing datasets.** In contrast to existing datasets, CMMA opens a door to sentiment, emotion, sarcasm, humor joint detection in multi-modal conversational context. In addition, it provides the relevance intensity across tasks, and for the first time incorporates the speaker's information (i.e., name, profession, sex, personality) and the conversational topics. Beyond the scope the multi-affection joint detection, CMMA can support a broad range of application fields, e.g, multi-modal deep learning.

## B.2 Multi-Modal Sentiment, Emotion, Sarcasm and Humor Detection

In this section, we focus on investigating related models covering multi-modal sentiment, emotion, sarcasm and humor.

**Sentiment and Emotion.** Numerous multi-modal sentiment and emotion analysis models have been proposed and evaluated on the above-mentioned datasets. For instance, Zhang et al. [6] presented a density matrix based textual and visual representation model for multi-modal sentiment analysis. Majumder et al. [24] presented the use of different RNNs to keep track of the emotional states, termed DialogueRNN, and achieved state-of-the-art performance. Li et al. [25] used the variational autoencoder (VAE) to extract the latent emotional features for multi-modal emotion recognition. Xu et al. [26] proposed a multi-modal attentive framework for their new task, which is aspect-based multi-modal sentiment analysis. Now, there have been few models that focus on simultaneously detecting sentiment and emotion. Akhtar et al. [27] proposed an inter-modal attention framework for simultaneously identifying sentiment and emotion. They attempted to model the inter-dependency across tasks, and proposed a multi-task learning model [28]. Khare et al. [29] used a cross-modal Transformer based multi-task framework to learn multi-modal embeddings.

**Sarcasm.** Utilizing multi-modal information for sarcasm detection has been explored recently. Cai et al. [15] designed a multi-modal sarcasm detection model in a hierarchical fusion manner. Pan et al. [30] designed a BERT architecture-based multi-modal model, which modeled both intra and inter-modality incongruity for sarcasm detection. Inspired by their work, Liang et al. [31] constructed a stacked graphical structure and updated multi-modal representation using graph convolutional networks (GCNs) for sarcasm detection. Liu et al. [32] proposed a quantum inspired multi-task learning model, which used quantum measurement to model the relationship between sentiment and sarcasm. Majumder et al. [33] proposed to use tensor product to model the interaction between sentiment and sarcasm. Zhao et al. [34] introduced a coupled attention networks (CANs) to incorporate both textual and visual clues for multi-modal sarcasm detection.

**Humor.** Although researchers have started to study humor in a multi-modal paradigm, there are only a limited number of models. Hasan et al. [35] presented a humor knowledge enriched Transformer (HKT) that could incorporated the humor knowledge into the cross-modal Transformer. Hazarika et al. [36] mapped multi-modal data into two subspaces to learn modality-invariant and modality-specific features for humor detection. Pramanick et al. [37] introduced multi-modal learning using optimal Transport (MuLOT), which employed self-attention and optimal transport to model inter- and cross-modal interaction for both sarcasm and humor detection. Bedi et al. [22] utilized the hierarchical attention and contextual attention mechanisms for Hindi-English sarcasm and humor detection. Both of them had neglected the correlation between sarcasm and humor.

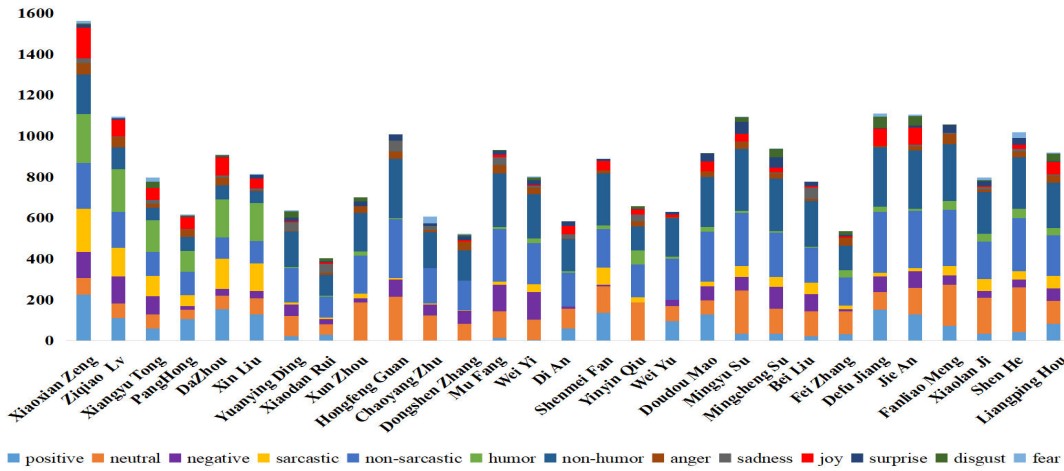

Figure 3: Speaker-level distribution per affection.

There is a gap in multi-task correlation modeling and multi-affection joint learning. CMMA will provide a solid foundation for the development of such communities. Beyond the scope the multi-affection joint detection, CMMA also benefits other applications, e.g, multi-modal deep learning.

## C. The Statistics of CMMA Dataset

In this work, eighteen famous Chinese TV series, e.g., "爱情公寓" (iPartment), "三国演义" (Romance of Three Kingdoms), "武林外传" (My Own Swordsman), etc., are chosen as our domain. Such TV series consist of multi-speaker conversations with utterances in forms of text (t), video (v) and audio (a), and cover different genres ($viz$. comedy, metropolitan opera, drama, crime thriller) and styles ($viz$. costume, war, idol, history, romance, family, crime). The statistics of the TV are shown in Table 1. We could notice that our domain includes six comedy shows, four metropolitan opera shows, five dramas and four thrillers, which is well-proportioned (6:4:5:4). Moreover, such TV shows cover various styles, e.g., costume, idol, romance, war, family, history, crime, which provide numerous conversation topics. Both actions will ensure us to collect balanced sentiment, emotion, sarcasm, humor, pride and love labels in the best possible way. To avoid copyright disputes, we have access to downloadable resources from public platforms. We argue that CMMA is made available for research purposes only. Any commercial use of this data is forbidden.

## D. The Inter-Agreement Comparison between CMMA and Benchmarking datasets

Sentiment/emotion annotation is a challenging task, in view of their intrinsic high-level abstraction and subjectivity. In the paper, we have elaborated on our efforts to enhance the reliability of the annotators. For example, (1) the annotators received professional guidance prior to annotation; (2) we made sure all the recruiters' uncertainties about the annotations were properly answered; (3) we provided annotation tips on the platform; (4) the annotators were instructed to annotate 100 examples, etc.

We also compare our inter-agreement (kappa score) with a range of benchmark datasets that reported the inter-agreement. As shown Tab. 2, we have the highest inter-agreement scores on all of eight annotation tasks. We also have recruited the highest number of annotators to enhance the quality. Note that MELD only asked the annotators to annotate the surprise as positive and negative sentiments, where their inter-agreement score will not be adopted to make comparison. UR-FUNNY/MHD/BBT/iSarcasm/Memotion did not report the inter-agreement. Sarcasm Corpus V2 and only reported the percentage agreement (80%), which was lower than ours (86.8%).

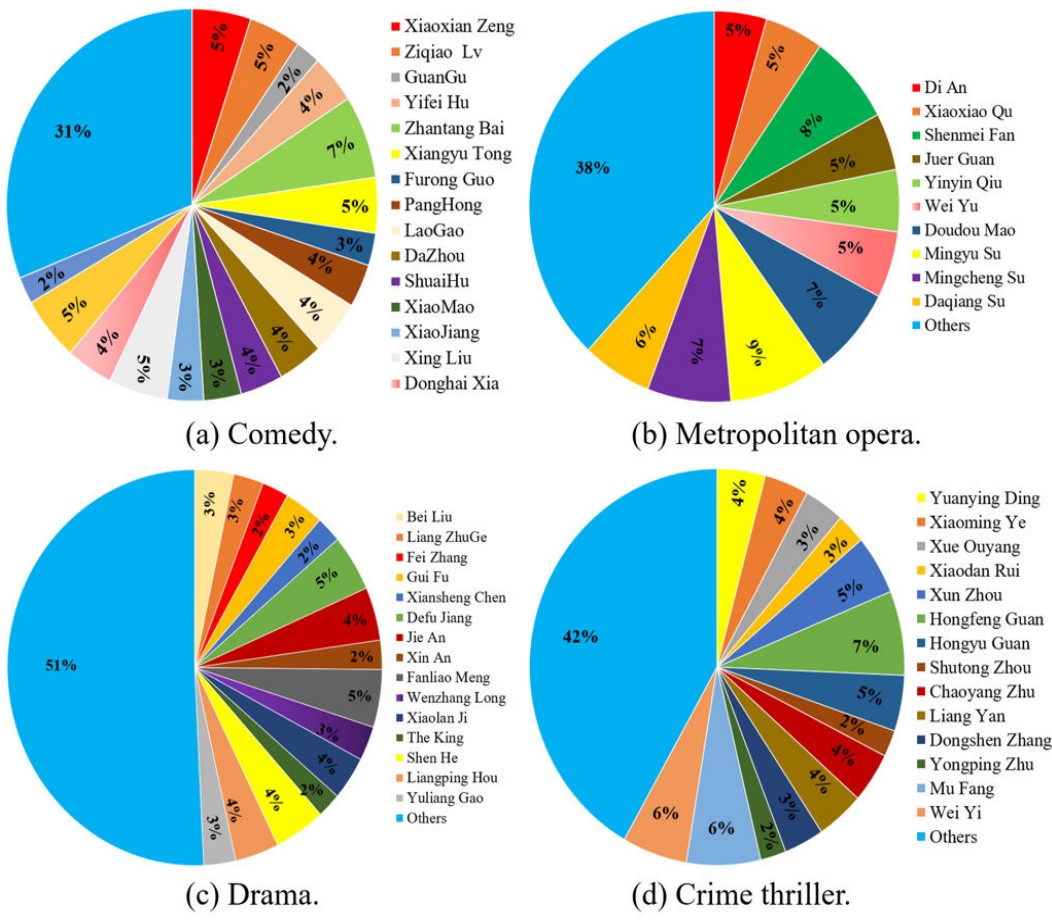

Figure 4: Overall speaker distribution.

## E. The Topic and Speaker Analysis

### E.1  How the Conversation Topic and Speaker Context Is Computed

(1) As for the topic description, we develop a brief template including time (if any), place, people, what happens and the ending, ask two volunteers to watch each conversation, and thus write their summarization following this template in one or two sentences. For example: *Alice came home and quarreled with Bob because Bob quite forgot her birthday today.* (2) As for the speaker information, these famous TV series have been recorded in Baidu Baike (which is the largest and most-read Chinese online encyclopedia). The leading characters' name, age, profession and their personality have also been detailed. We use those descriptions as the speaker's information. The topic and speaker information are represented by PLMs and merge with the input. (3) Affection detection model targets at simulating and understanding humans, where humans' personality indeed influences their affective states. The introduction of the speaker information will deepen the understanding of human affection, and help improve the performance. In addition, we can leverage the bias mitigation approach (e.g., the double hard bias mitigation) to remove the negative effect of the implicit bias and keep the beneficial information.

### E.2  The Speaker Distribution

We will only study the distribution of leading characters since there are about 500 speakers in CMMA dataset. Fig. 3 illustrates the speaker-level distribution per affect. We observe that the positive, joy, sarcastic and humor affections are more likely happen in the characters of sitcoms, e.g., Xiaoxian Zeng, LaoGao, Yifei Hu, etc. These characters produce plenty of professions related punchlines.

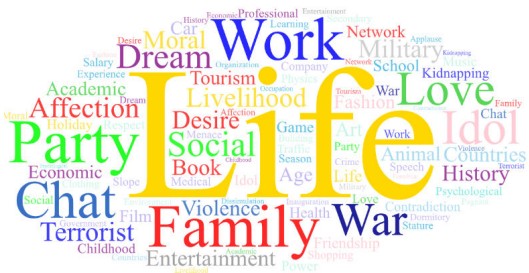

Figure 5: Topics of the conversations in CMMA dataset.

Anger and fear emotions occur more frequently in the characters of drama and crime thriller, e,g, Chaoyang Zhu, Hongyu Guan, Shenmei Fan, etc. These speakers rarely express positive or joy emotions, since they are tragic figures in the TV shows. This implicates that the speaker's information is also valuable for affection judgment. Neutral, non-sarcastic and non-humor affections occur in every character for all TV shows.

Fig. 4 shows the overall coverage of the major speakers across the dataset. To satisfy the requirement of "representativeness", we illustrate the major speakers for four genres ($viz.$ comedy, metropolitan opera, drama, crime thriller) respectively. We notice that they are evenly distributed, which has avoided the speaker bias of such popular speakers. In addition, such speakers account for more than half of the total distribution, which is expected since they play affective role. Multiple infrequent supporting speakers (< 1% utterances) are grouped as Others.

## F. Model Building

### F.1 Text Encoder

We use GloVe 6B to initialize the 100 dimensional word embeddings as inputs for DCNN and BiLSTM. As for BERT, the dimension is 768.

**BiLSTM.** It consists of two LSTM layers that read the input sequence forwardly and backwardly to generate a series of bidirectional hidden states. The $i^{th}$ hidden representation is obtained by merging the bidirectional hidden states, e.g., $\boldsymbol{h}_i = \overrightarrow{\boldsymbol{h}}_i \parallel \overleftarrow{\boldsymbol{h}}_i$, where $i \in [1, 2, ..., n]$. In BiLSTM, the dimensions of forward and backward hidden states are set to 50 respectively. Finally, the final hidden sate $\boldsymbol{h}_n$ is used as the sequence representation.

**BERT.** We fine-tuned the BERT-base including 12 layers and 110M parameters as the text encoder. Each sequence will be padded or truncated to the size of 50 before it is input. The obtained representation of the first token in the sequence (i.e., the [CLS] token) is used as the output of the encoder, where the dimension is 768.

**GPT-2.** We adopt GPT-2 including12 layers and 117M parameters as the text encoder. Each sequence is limited to the size of 50 before it to be input. The text representation is treated as the output of the encoder, where the dimension is 768.

**GPT-3.** We adopt GPT-3 API to perform emotion recognition.

### F.2 Video Encoder

**Efficient.** The size of the input images is $408 \times 612 \times 3$. The first convolutional layer has 96 kernels of size $12 \times 40 \times 3$ with a stride of 4 pixels. The second convolutional layer has 256 kernels of size $5 \times 5 \times 96$ with a stride of 2 pixels. The third convolutional layer has 384 kernels of size $3 \times 3 \times 256$. The forth convolutional layer has 384 kernels of size $3 \times 3 \times 384$, and the fifth convolutional layer has 256 kernels of size $3 \times 3 \times 384$.

**ResNet.** The ResNet18 pre-trained model is used in our experiments. All the images are resized to $612 \times 612 \times 3$ before they are feed into the model.

Table 4: Dataset statistics.

| | | CMMA | | |
| | | Train | Validation | Test |
|---|---|---|---|---|
| Sentiment | Positive | 1889 | 559 | 923 |
| | Neutral | 9324 | 2694 | 2249 |
| | Negative | 2575 | 793 | 789 |
| Emotion | Anger | 1610 | 375 | 382 |
| | Sadness | 620 | 168 | 99 |
| | Joy | 1950 | 406 | 182 |
| | Surprise | 534 | 514 | 371 |
| | Disgust | 821 | 118 | 19 |
| | Fear | 477 | 92 | 43 |
| | Neutral | 7776 | 2373 | 2865 |
| Sarcasm | Sarcastic | 2172 | 525 | 446 |
| | Non-sarcastic | 11616 | 3521 | 3515 |
| Humor | Humor | 1368 | 602 | 619 |
| | Non-humor | 12420 | 3444 | 3342 |
| Pride | Pride | 537 | 566 | 800 |
| | Non-pride | 13251 | 3480 | 3161 |
| Love | Immediate | 46 | 13 | 23 |
| | Growing | 217 | 32 | 49 |
| | Empty | 11 | 10 | 10 |
| | None | 13514 | 3991 | 3879 |

## F.3 Audio Encoder

**VGGish.** VGGish resamples the audio to 16kHz monophonic audio, obtains the spectrum features by using the Fourier transform, and calculates the MEL spectrum in the 64th-order filter. Such features are framed with 0.96s, and a 128-dimensional feature vector is finally obtained.

## F.4 The Description of Relevance-Aware Model

In this work, we only incorporate the relevance intensity across affections as extra knowledge into model training. For our relevance-aware models (RaM), we forward the tri-modal fused features through six separate (i.e., task specific) dense layers and obtain six output feature vectors. Such feature vectors are treated as parts of input and prepared for the subsequent tasks of six affections detection. Then, we start to investigate the relevance intensity of the target utterance. For example, if the $emotion \rightarrow sentiment$ relevance intensity is -2, the output vectors of the sentiment dense layer will be directly added to the feature vectors of the emotion dense layer to build a new input for emotion detection. If the $emotion \rightarrow sentiment$ relevance intensity is -1, the output features of the sentiment dense layer will be halved before adding with the output of emotion dense layer. If the $emotion \rightarrow sentiment$ intensity is 0, the sentiment dense layer will not be added to the emotion layer. This action naturally leverages the knowledge from other affections in an relevance-aware manner.

## G. Model Training

Table 4 shows the dataset statistics. We use Pytorch to build all models. To avoid overfitting, we choose to perform early stopping during training. During training, the learning rate is set to $1 \times 10^{-2}$ and the epoch is 60 if the encoder includes pre-trained model, otherwise they are set to $1 \times 10^{-3}$ and 100 respectively. The dropout rate in the model is 0.5. In our models, cross entropy with $L2$ regularization is used as the loss function, as shown in Eq. 1:

$$\zeta_s = -\sum_i \sum_k y_{k,s}^i log\hat{y}_{k,s}^i + \tau_r \|\phi\|^2$$

$$\zeta = \sum_{j=1}^{6} w_j \zeta_j$$

(1)

where $\zeta_s \in \{\zeta_{sen}, \zeta_{emo}, \zeta_{sar}, \zeta_{hum}, \zeta_{prd}, \zeta_{lov}\}$, $y_{k,s}^i$ denotes the ground truth of the $k^{th}$ utterance for the $i^{th}$ conversation, $\hat{y}_{k,s}^i$ is the predicted distribution. $\tau_r$ is the coefficient for $L2$ regularization. As

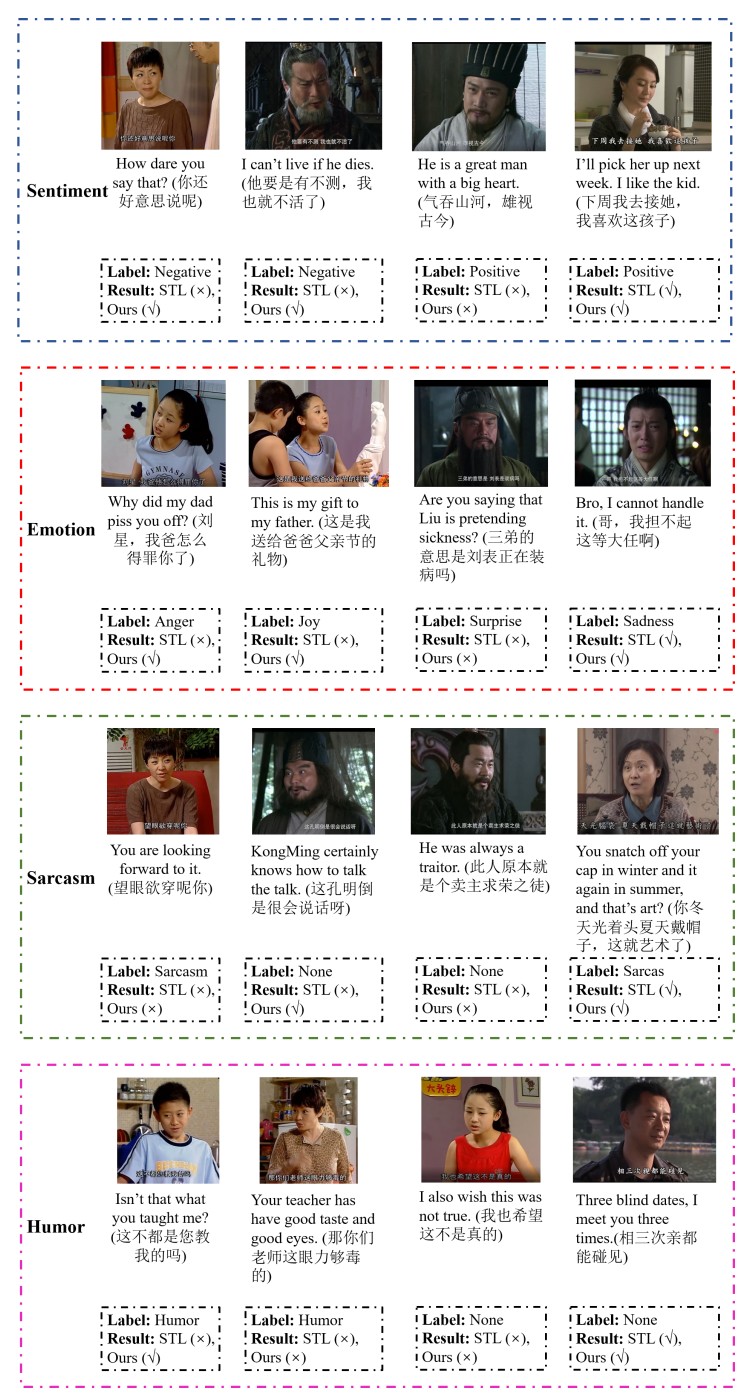

Figure 6: Wrongly classified multi-modal samples where MTL framework performs better than the STL framework.

for optimizer, we choose Adam to optimize the loss function. We use the back propagation method to compute the gradients and update all the parameters. It takes about 70 minutes for the state-of-the-art system (i.e., BERT+ResNet+VGGish) to train its best performance over CMMA via 1×RTX A6000 GPU. We run the experiments using five-fold cross-validation on all models.

Table 5: Error distribution.

| Setup | Sentiment | | | Emotion | | | | | | | Sarcasm | | Humor | |
|---|---|---|---|---|---|---|---|---|---|---|---|---|---|---|
| | Pos. | Neg. | Neu. | Ang. | Sad | Joy. | Sur. | Dis. | Fear | Neu. | Sar. | Non-Sar. | Hum. | NonHum. |
| Only STL fails | 82 | 126 | 276 | 54 | 13 | 10 | 35 | 2 | 5 | 419 | 24 | 168 | 39 | 163 |
| Both models fail | 152 | 104 | 318 | 42 | 8 | 4 | 91 | 9 | 4 | 382 | 113 | 362 | 107 | 389 |

Table 6: Human evaluation vs multi-affective analysis model.

| Method | Sentiment | Emotion | Sarcasm | Humor |
|---|---|---|---|---|
| Annotator 1 | 87.15 | 83.48 | 91.74 | 94.03 |
| Annotator 2 | 88.07 | 80.27 | 90.36 | 91.28 |
| Annotator 3 | 82.56 | 85.82 | 94.86 | 90.82 |
| Avg. | 85.92 | 83.19 | 92.32 | 91.81 |
| BERT+ResNet+VGGish | 73.76 | 68.68 | 80.65 | 77.58 |

## H. Error Analysis and Human Evalulation

In order to explore the potential of the state-of-the-art baseline, we perform an error analysis and show a few misclassification cases (utterance plus image), including the cases that our MTL setup predicts correctly while STL fails, that both setups predict correctly and that both setups fails to predict correctly. These cases are shown in Fig. 6. We notice that misclassification for BERT+ResNet+VGGish often happens in five categories of samples, i.e., positive, surprise, sarcasm and humor.

For sentiment classification, we see that misclassification for STL framework often happens in the situation where the literal meaning of an subjective expression is difficult to understand, such as short sentence, classical style of writing, etc. Through utilizing the humor and emotion knowledge, MTL framework makes the correct prediction and obtains a significant improvement. However, we observe that MTL might also struggle in understanding classical Chinese words.

For emotion recognition, we see that both MTL and STL frameworks fail in understanding surprise emotion. Because surprise is a more abstract human emotion, which can be expressed with a positive or negative sentiment. The shared knowledge helps few over this task.

For sarcasm detection, we notice that misclassification for STL framework often happens in the situation where the literal meaning of an ironic expression differs from its real sentiment. We observe that MTL might struggle in intricate cases requiring external information, e,g., the speaker's character. For humor detection, we observe that both our MTL setup and STL fail in the similar situation.

In addition, we here present the distribution of misclassification cases for sentiment, emotion, sarcasm and humor tasks, as shown in the Tab. 5 The distribution shows that MTL gains better performance for negative, anger, sad, joy, fear classes. Both STL and MTL often fail in five categories of samples, i.e., positive, neutral, surprise, disgust, sarcastic and humor. These results stay in step with our error analysis in the following descriptions.

**Comparison to Human Evaluation.** We create a new test set (following the same rules of CMMA) including 40 conversations with 218 multi-modal utterances, and recruit three volunteers to evaluate the sentiment, emotion, sarcasm and humor labels. The human labels[3] are compared with our best setting, BERT+ResNet+VGGish. Table 6 presents the comparative results. It can be seen that our model under-performs human performance by a huge gap (16.5%, 21.1%, 14.5%, 19.0% ). Therefore multi-modal multi-affect joint detection is a challenging task that requires innovations in theory and practice. The proposed CMMA dataset will serve as a benchmark for evaluation.

## I. Further Comparison between STL and RaM

The crucial role of cross-affection correlation in affective analysis has been widely discussed in literature. The fact that multiple affections are tangled and their interaction and correlation have been demonstrated in numerous studies in NLP, Psychology and Philosophy.

---

[3]The human annotations have a high inter-annotator agreement, 0.77, 0.67, 0.69 and 0.81 for the four tasks.

Table 7: The comparison between STL and RaM on the new test set.

| Setup | Sentiment | Emotion | Sarcasm | Humor |
|-------|-----------|---------|---------|-------|
|       | Ma-F1     | Ma-F1   | Ma-F1   | Ma-F1 |
| STL   | 70.03     | 57.89   | 72.55   | 71.67 |
| RaM   | 73.77     | 59.22   | 74.78   | 74.74 |
| %     | 5.34      | 2.15    | 3.07    | 4.28  |

We have empirically shown in Tab. 9 (in the main paper) that cross-affection correlation can boost the detection of each single affection on our newly constructed dataset. With a relevance-aware joint detection model (RaM) that incorporates cross-affection correlations, we attain a significant improvement of 4.35%, 3.36%, 4.53%, 3.70% in F1 over the single task learning (STL) model.

A potential threat to the validity of the argument is that when an annotator judges different affections simultaneously, the judgments may influence each other and produce a bias to the annotation result. To rule out this bias, we create a new test set including 20 conversations and 132 multimodal utterances, and evenly split them into four subsets. Four volunteers are recruited to annotate the sentiment, emotion, sarcasm and humor labels for each subset respectively. In this way, each utterance has a single affective label annotated by a single volunteer. Finally, the results also show that RaM beats STL by a large margin (see Tab. 7). The argument strengthens the argument that multi-task correlation has enormous potential for affection detection, and illustrates that our dataset is suitable as a benchmark for studying the benefit of cross-affection correlation to the detection of multiple affections.

## J. Discussion of the Application

We propose SIX multi-modal affection detection tasks for our dataset, which covers a wide range of applications in multi-modal affection analysis. Our dataset includes more tasks than any other benchmark dataset, i.e., 6 tasks (ours) v/s 2 3 task (i.e., MUMOR), v/s 1 2 tasks (MELD, IEMOCAP, MUStARD, ScenarioSA, etc).

We report the performance of strong baselines on our dataset. We do not aim at creating a new SOTA on this dataset, but at demonstrating the quality of the collected data and identifying the challenges. The experiments have showcased that the affection analysis is influenced by many factors, including each modality, conversational context, and more importantly cross-affection correlations. On the other hand, there is still a huge gap between the strong baselines and human performance (Tab. 7), and poses a challenge for leveraging all these factors to improve affection detection.

## K. Explanation of Our Motivation of Pride and Love Annotation

In Psychology and Philosophy, pride and love are kinds of primitive instinct and complex sentiments for strongly expressing a sense of self. Both of them involve not only the linguistic expression, but also the cognitive mind driving human sentiments and emotions. Hence, we argue that there are close relations between pride/love and sentiment, where pride/love stealthily dominates sentiment while sentiment also has had effects on them. For example, "Hey guys, you are going to love him, his smile is so gentle and warm. Cute boy." expresses the speaker's love and positive sentiment.

Given the importance of pride/love in sentiment and emotion understanding, researchers are a step closer to realize general artificial intelligence if a machine is able to find a deeper understanding of human pride/love or even make reasonable pride/love-aware responses. However, it is still a fascinatingly understudied new task in NLP and multi-modal affective computing. In addition, automatic pride/love understanding is also a challenging task. For instance, (a) the lack of publicly available datasets; (b) how to define and extract pride/love knowledge?

Motivated by this, we make the first attempt and introduce two new sentiments, i.e., Pride and Love, to deepen the understanding of sentiment. We also design two fresh tasks to show their potential.

Table 8: Pride and Love task results of different models.

| Model | Text | Video | Audio | Pride P | Pride R | Pride $M_a$-F1 | Love P | Love R | Love $M_a$-F1 |
|---|---|---|---|---|---|---|---|---|---|
| Text | BiLSTM | - | - | 47.26 | 52.36 | 49.68 | 42.25 | 44.78 | 43.48 |
|  | BERT | - | - | 47.72 | 57.89 | 52.31 | 41.24 | 52.06 | 44.78 |
|  | GPT-2 | - | - | 44.19 | 47.77 | 45.91 | 41.16 | 59.77 | 46.05 |
|  | GPT-3 | - | - | 46.45 | 48.55 | 47.48 | 47.41 | 49.36 | 48.36 |
| Video | - | EfficientNet | - | 40.99 | 49.39 | 44.80 | 33.18 | 43.66 | 37.7 |
|  | - | ResNet | - | 49.18 | 52.36 | 50.72 | 44.96 | 53.38 | 47.29 |
| Audio | - | - | VGGish | 37.86 | 42.11 | 39.87 | 33.26 | 44.26 | 35.54 |
| Text+Video | BiLSTM | EfficientNet | - | 47.65 | 56.01 | 51.49 | 45.93 | 62.20 | 51.43 |
|  | BiLSTM | ResNet | - | 49.65 | 56.68 | 52.93 | 47.69 | 53.39 | 49.49 |
|  | BERT | EfficientNet | - | 64.88 | 60.32 | 62.52 | 49.97 | 62.02 | 53.29 |
|  | BERT | ResNet | - | 69.21 | 64.64 | 66.85 | 58.01 | 70.57 | 61.47 |
|  | GPT-2 | EfficientNet | - | 53.57 | 55.62 | 54.57 | 56.98 | 60.55 | 55.83 |
|  | GPT-2 | ResNet | - | 61.15 | 57.35 | 59.19 | 58.02 | 65.16 | 58.57 |
| Video+Audio |  | EfficientNet | VGGish | 41.34 | 53.85 | 46.78 | 51.67 | 57.61 | 53.03 |
|  | - | ResNet | VGGish | 43.82 | 54.52 | 48.59 | 50.96 | 68.86 | 57.34 |
| Text+Audio | BiLSTM | - | VGGish | 46.27 | 51.82 | 48.89 | 51.73 | 65.16 | 52.07 |
|  | BERT | - | VGGish | 55.82 | 55.60 | 55.71 | 50.87 | 65.56 | 56.25 |
|  | GPT-2 | - | VGGish | 49.81 | 53.98 | 51.81 | 47.98 | 64.12 | 53.18 |
| Text+Video+Audio | BERT | EfficientNet | VGGish | 72.01 | 69.10 | 70.52 | 60.16 | 74.03 | 66.38 |
|  | BERT | ResNet | VGGish | 75.58 | 79.35 | 77.42 | 67.18 | 75.67 | 70.31 |
|  | GPT-2 | EfficientNet | VGGish | 68.24 | 70.45 | 69.32 | 52.84 | 63.93 | 54.59 |
|  | GPT-2 | ResNet | VGGish | 75.59 | 73.55 | 74.56 | 67.23 | 73.17 | 68.89 |
| Trimodal *vs* Bimodal (%) | - | - | - | +9 | +22.7 | +15.8 | +15.8 | +7.2 | +14.2 |

Due to their complexity, prior to annotation of pride and love, the annotators received professional guidance covering the concept of pride and love, the criteria for labeling and negative examples, etc. A positive pride example: *I help you get the lead actress!*

In Table 8, we also report the performance results of different models on Pride and Love tasks.

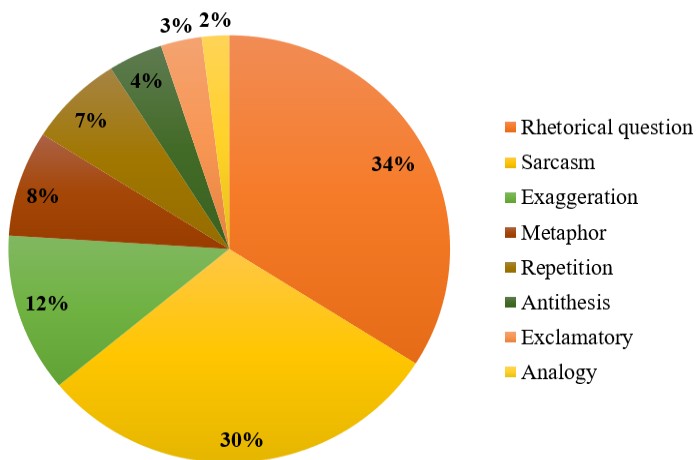

Figure 7: The distribution of rhetorical devices.

## L. Linguistic Insights

We acknowledge the importance of linguistic insight in understanding the multi-modal and multi-affect nature of human communication. We randomly sample 500 conversations, and analyze the usage of the emotional words and the distribution of rhetorical devices. The results are shown in Table 9.

We have listed 30 typical emotional words and given their percentages. For example, the percentage of "anger" is computed by dividing the number of the utterances containing angry words into the total number of utterances. Note that the sum of their percentages is not one due to the existence of neutral utterances. We can notice that the distribution of emotional terms is consistent with the distribution of emotion labels. For example, the percentage of joy words is 10.1%, where the percentage of joy labels is 11.6%. Fear words accounts for 2.2%, which is consistent with the percentage of fear labels (2.9%). This result shows that emotional words could contribute to the judgement of emotion.

In addition, this can more comprehensively reflect the rich emotional connotations contained in the dataset. Such words are used to express various affects of the speakers and enhance the infectivity of the language. In addition, a few words are also used for sarcasm and humor, such as using positive terms like "happiness" and "surprise" to express irony. The dataset employs a wide variety of rich emotion words, including words like anger, sadness, fear, and joy, which reflects the abundant emotional nuances contained in the samples. The distribution of emotional terms are consistent with the distribution of emotion labels and hence could contribute to the judgement of emotion

In addition, the distribution of rhetorical devices is shown in Fig. 7. It can be seen that the usage frequency of rhetorical question is the highest, accounting for 34% together, with flexible and vivid tones; sarcasm and exaggeration are very common. The flexible use of various rhetorical devices makes the language both expressive and humorous and mocking. We see that such linguistic features enhances the artistry of the language, making it more infectious and interesting. In one word, CMMA is a corpus with distinct language styles and vivid emotional expression.

Table 9: The percentage of the emotional words.

| Affect | Words | Percentage(%) |
|---|---|---|
| Anger | anger, disgust, indignation, annoyance, irritability, fury, etc. | 8.3 |
| Sadness | sadness, sorrow, disappointment, depression, heartbreak, regret, etc. | 2.6 |
| Fear | fear, scare, worry, timidity, cowardice, horror, etc. | 2.2 |
| Joy | happiness, pleasure, surprise, excitement, delight, ecstasy, etc. | 10.1 |
| Others | anxiety, tension, jealousy, love, sympathy, belief | 9.3 |

## M. Explanations of Our Scripted Data Source

We understand that the potential biases may arise when sourcing content from scripted television shows. But we have to say that TV shows can still serve as a valuable resource for affect detection research, due to three reasons:

(1) The perceived emotions of actors are not purely fabricated, but rather portrayals of genuine human experiences and reactions. Though likely more exaggerated than real world expressions, by having nine trained annotators label affects based on perception rather than intentions, we can collect labels that capture the essence of displayed affect. (2) TV content is easy to obtain, has no privacy concerns, and can be openly published and reused. Real conversation collection requires obtaining a large amount of recordings or texts from actual dialogue situations. This process involves privacy concerns and legal compliance issues. (3) TV show dialogues cover a wide range of language styles and affect expressions, allowing collection of richly annotated training samples.

Of course, we understand the importance of addressing biases. To mitigate these concerns, we have made several efforts:

(1) Data Diversity: We took great care in selecting a diverse range of TV series with various subjects and styles. By incorporating a wide variety of contexts, characters, and scenarios, our dataset, CMMA, aims to capture a more representative sample of human emotions in conversational settings. (2) Manual Annotation: The affect labels in CMMA were manually annotated by nine well trained experts. This process helps ensure that the affects captured in the dataset are based on genuine expressions and helps minimize subjectivity and ensure accuracy. (3) Robust Analysis: In our experiments, we have performed rigorous analysis. We have also discussed the limitations and provide transparency about the dataset's scope to ensure its appropriate usage.

Despite the potential limitations, we firmly believe CMMA offers valuable insights into the multi-modal and multi-affection nature of human communication. Our work seeks to lay a strong foundation for further research in human affect understanding.