# OpenReview forum: "CMMA: Benchmarking Multi-Affection Detection in Chinese Multi-Modal Conversations"
_NeurIPS.cc/2023/Track/Datasets_and_Benchmarks — NeurIPS 2023 Datasets and Benchmarks Poster_

### Official Review · Reviewer_LR2c · 2023-06-29
**Review of Submission-448**

**Rating:** 6
**Confidence:** 4
**Clarity:** The paper is well-written, but some o…

**Strengths:**

- The proposed dataset encompasses the most comprehensive aspects for conversational affection extraction.
- The authors have taken care to provide a detailed description of the data collection and processing specifications, ensuring the quality and integrity of the data.
- Notably, the annotation process involved seven annotators for each sample, and the satisfactory agreement score indicates a high level of consistency in the annotation.


**Additional Feedback:**

See the section “Opportunities For Improvement”.

**Correctness:**

The authors have diligently provided a comprehensive explanation of their data collection and processing specifications, highlighting the significance of adhering to relevant regulations and ethical requirements.
The benchmark and experimental results effectively demonstrate the challenging nature of the task and the benefits of employing Multi-Task Learning (MTL) for affective extraction.
Based on these details, the claims made in the submission appear to be correct.



**Documentation:**

The authors have provided a comprehensive description of the data collection and annotation processes.
The URL for accessing the dataset is made available to the reviewers.
However, there is a missing provision for hosting, licensing, and a maintenance plan.


**Limitations:**

The authors mentioned a potential limitation of the dataset, which is the possibility of introducing bias.
In addition to that, there are other limitations worth considering, as mentioned above.


**Opportunities For Improvement:**

The dataset's exclusive availability in Chinese may limit its applicability in non-Chinese-speaking contexts or for researchers working with languages other than Chinese.

The primary objective of the proposed task is to explore the potential benefits of understanding one affection type for the understanding of another.
However, the model presented in this paper solely employs multi-task learning without incorporating additional interactions between different affective aspects. Consequently, the obtained results may not fully reflect the comprehensive usefulness of different labels in aiding each other's extraction. It is important to note that improvements in the task can also be achieved by considering the statistical co-occurrence of the labels.

Additionally, the number of turns in the dataset is relatively low, with an average of only around 3 turns. This limitation may not fully reflect the complexities of real conversational environments.


**Relation To Prior Work:**

The authors have provided a comprehensive comparison of their work with previous research and highlighted the contribution.

**Summary And Contributions:**

The paper presents the CMMA dataset, a multi-modal dataset that includes text, video, and audio information.
The dataset is manually annotated, and it records various aspects such as emotion, sentiment, sarcasm, humor, pride, and love.
The paper demonstrates the potential of the dataset as a benchmark for studying the benefit of cross-affection correlation to the detection of multiple affections.

---

> ### Author Response · Authors · 2023-08-10
>
> $Q1$: The dataset’s exclusive availability in Chinese may limit its applicability in non-Chinese-speaking contexts or for researchers working with languages other than Chinese.
>
> $R1$:  Thank you for providing valuable feedback on our paper. We acknowledge that our scope is limited to the Chinese context. But we would like to emphasize the importance of our dataset to the affective computing community and its potential significance in promoting multi-language and multi-affect research. Although CMMA is currently available only in Chinese, the affect labels and conversational features it contains have universal and transferable properties. This indicates the potential value of CMMA for cross-lingual research. In addition, we state that the experience of constructing multi-affect dataset could be used for collecting/annotating/benchmarking English dialogues.
>
> $Q2$: The primary objective of the proposed task is to explore the potential benefits of understanding one affection type for the understanding of another. However, the model presented in this paper solely employs multi-task learning without incorporating additional interactions between different affective aspects. Consequently, the obtained results may not fully reflect the comprehensive usefulness of different labels in aiding each other's extraction. It is important to note that improvements in the task can also be achieved by considering the statistical co-occurrence of the labels.
>
> $R2$: We sincerely appreciate your concerns and feedback. We would like to provide further insights into the proposed multi-task learning approach that we have indeed incorporated interactions between different affective knowledge.
>
> In our research, we introduce the relevance intensity between different affects and sentiments as extra knowledge and utilize our relevance-aware models (RaM) to jointly detect sentiment-emotion and sarcasm-humor pairs. By employing the relevance-aware mechanism in model training, we take into account the interplay between different affects to achieve a more comprehensive affection detection.
>
> We emphasize that we explicitly capture the directional affective interactions in our model, which is fundamentally different from relying on the non-directional statistical co-occurrence of the labels. Specifically, we forward the tri-modal fused features through six separate (i.e., task-specific) dense layers, obtaining six output feature vectors. Then, we investigate the relevance intensity of the target utterance. For instance, if the emotion → sentiment relevance intensity is 2, the output vectors of the emotion dense layer will be directly added to the feature vectors of the sentiment dense layer to build a new input for sentiment classification. If the emotion → sentiment relevance intensity is -1, the output features of the sentiment dense layer will be halved before being added to the emotion branch. This approach naturally leverages the shared knowledge from other affects. This allows features of one affect to directly act on another, reflecting their inherent dependencies. Our relevance-aware multi-task learning approach goes beyond traditional multi-task learning by explicitly leveraging the relevance intensity values to dynamically adjust the strength of interaction between affects.
>
> Thank you for the valuable suggestion. We will also propose a more refined and explicit multi-affect interaction modeling approach in the future.
>
>
> $Q3$: Additionally, the number of turns in the dataset is relatively low, with an average of only around 3 turns. This limitation may not fully reflect the complexities of real conversational environments.
>
> $R3$: Thank you for your thoughtful feedback. We appreciate your observation regarding the number of turns in our dataset.
>
> The turns of a conversation on social media platforms can vary due to factors like platform, topic, participants, etc. (usually 2~10 turns). The literature review shows that the average values of turns of the current dialogue datasets range from 2 to 7 [1-3]. Hence, our dataset falls within a reasonable range. Furthermore, since our dataset consists of multi-party conversations, e.g., a 3-turns dialogue would normally include about 9 utterances.
>
> $\bullet$[1] EVA: An Open-Domain Chinese Dialogue System with Large-Scale Generative Pre-Training, 2021 [Average turns: 2.1]
>
> $\bullet$[2] MEmoR: A Dataset for Multimodal Emotion Reasoning in Videos, ACM MM, 2020 [Average turns: 1.3]
>
> $\bullet$[3] DailyDialog: A Manually Labelled Multi-turn Dialogue Dataset, IJCNLP 2017 [Average turns: 7.9]
>
>
> $Q4$: In addition to that, there are other limitations worth considering, as mentioned above.
>
> $R4$: Thank you for your thoughtful feedback. We have added more descriptions of other limitations (in the revised version, Sec.5, Page 10).

---

### Official Review · Reviewer_hf9n · 2023-07-18
**Versatile multiomodal affective science dataset**

**Rating:** 8
**Confidence:** 4
**Clarity:** The paper is well written and well or…

**Strengths:**

Data are rich and multi-faceted. For example, both affect in valence and arousal terms, as well as individual discrete emotions, such as anger or joy, are captured, in addition to several other aspects of affect and emotion that are less commonly available. The strength of the dataset lies in providing these data together, for the same dataset. I can see this versatile dataset offering several exciting opportunities for further studies.

**Additional Feedback:**

This dataset might be of potential interest to linguists, in addition to affective science research, as the tagging of sarcasm and humor might provide further insight into the relationship of language use and affect in relation to character/demographic traits.

**Correctness:**

The dataset is constructed soundly, and thoroughly tested through various methods, including human evaluation and statistical testing.

**Documentation:**

The dataset will be made publicly available, which means it can be accessed by others and used for further investigation. Ratings seem straightforward in their description in the paper, as well as easy to find and navigate in the repo. I did find that the authors failed to conceal their identity in the anonymous repo. Some documents have chinese character titles, which might make it hard for non-chinese speakers to understand the function of these documents, or programs to read out these files.

**Ethics:**

Authors did fail to conceal their identity in githup repo https://github.com/annoymity2022/Chinese-Dataset/blob/main/Datacard/Datacards.pdf

**Limitations:**

Limitations described in the paper are brief and could be expanded to include details on potential biases, limitations to generalizations, and others. One additional point would be that the data are collected from Chinese TV-shows, so they reflect scripted communicative exchanges.

**Opportunities For Improvement:**

The work would greatly benefit from revised figures. The figures are too small, warped, and illegible, as none of them are vector graphics. I recommend to either include all figures as vector graphics, or reduce the number of figures and increase the size for the remaining ones. Another aspect that was unclear to me is the terminology and function ‘forward’ and ‘backward’ correlations offered on a numeric scale. This aspect seemed under described and -interpreted, and requires further exploration in a revised version of the submission.

**Relation To Prior Work:**

Yes, although the review of prior work is brief, it is adequate and differences are clearly described.

**Summary And Contributions:**

The present submission poses a significant contribution to the field, by introducing a new dataset of Chinese TV series, annotated for several affective dimensions simultaneously. Benchmarking for the dataset, as well as potential relationships between the assessed dimensions are discussed. This work promises a rich dataset of based on Non-English samples and offers a variety of interesting investigative opportunities to researchers from several disciplines.

---

> ### Author Response · Authors · 2023-08-10
>
> $Q1$: The work would greatly benefit from revised figures.
>
> $R1$: We have revised all the figures and move one Figure to Appendix to make them clearer (in the revised version).
>
> $Q2$: the terminology and function ‘forward’ and ‘backward’ correlations.
>
> $R2$: Thanks for your valuable feedback on our work. We prefer to annotate the one association that has the greatest contribution to help annotate the other task. The ‘forward' correlation indicates the degree to which one affection contributes to the understanding of another affection. The “backward” indicates the contrary contribution. For example, if the annotator thinks that the sentiment contributes more the emotion than emotion contributes to sentiment, then he will give a positive relevance score when the main direction is sentiment  emotion. The reason is that such a choice will help determine the main-secondary task and balance high quality affection resources and the cost.
>
> $Q3$: Limitations described are brief and could be expanded to include details on potential biases, limitations to generalizations, and others.
>
> $R3$: We appreciate your suggestion to expand on the limitations described in the paper. In response to your feedback, we have revised the limitations section (Sec.5, Page 10).
> (1) Potential Biases: We acknowledge that our dataset, CMMA, was collected from TV shows, which could introduce certain biases in emotion expressions and communication styles. The portrayal of emotions by actors in scripted scenes may not fully represent real-life social interactions. Additionally, cultural and regional differences depicted in TV content may influence emotion expressions, potentially affecting the dataset's generalizability to diverse sociolinguistic contexts.
> (2) Limitations to Generalizations: As with any dataset, the CMMA dataset also has its unique characteristics and limitations that may affect the generalization of our results. The dataset’s focus on Chinese conversations from television shows may limit the direct application of our findings to other languages or real-life social contexts. We encourage future researchers to consider cross-linguistic validation and collect diverse datasets to ensure broader generalization.
>
> $Q4$: The data are collected from Chinese TV-shows, so they reflect scripted communicative exchanges.
>
> $R4$: We appreciate your concern regarding the potential biases that may arise when sourcing content from scripted shows. But we have to say that TV shows can still serve as a valuable resource for affect detection research, due to three reasons:
>     (1) The perceived emotions of actors are not purely fabricated, but rather portrayals of genuine human experiences and reactions. Though likely more exaggerated than real world expressions, by having NINE trained annotators label affects based on perception rather than intentions, we can collect labels that capture the essence of displayed affect.
>     (2) TV content is easy to obtain, has no privacy concerns, and can be openly published and reused. Real conversation collection requires obtaining a large amount of recordings or texts from actual dialogue situations. This process involves privacy concerns and legal compliance issues.
>     (3) TV show dialogues cover a wide range of language styles and affect expressions, allowing collection of richly annotated training samples.
>
> To mitigate these concerns, we have made several efforts:
> (1) Data Diversity: We took great care in selecting a diverse range of TV series with various subjects and styles. By incorporating a wide variety of contexts, characters, and scenarios, our dataset, CMMA, aims to capture a more representative sample of human emotions in conversational settings.
> (2) Manual Annotation: The affect labels in CMMA were manually annotated by Nine well trained experts. This process helps ensure that the affects captured in the dataset are based on genuine expressions and helps minimize subjectivity and ensure accuracy.
> (3) Robust Analysis: In our experiments, we have performed rigorous analysis. We have also discussed the limitations and provide transparency about the dataset’s scope to ensure its appropriate usage (Sec. 5, Page 10).
>
> Despite the potential limitations, we firmly believe CMMA offers valuable insights into the multi-modal and multi-affection nature of human communication. Our work seeks to lay a strong foundation for further research in human affect understanding. And we have added the detailed explanations in Appendix M in Supplementary document (Page 15).
>
> $Q5$: Authors did fail to conceal their identity in githup repo.
>
> $R5$:  Thanks for your constructive feedback. NeurIPS DB track states that “Authors can choose to submit either single-blind or double-blind”. So we ignored hiding our identity in the data card in view that single-blind submissions are allowed. To rectify this issue, we have promptly made the necessary updates to the GitHub repository and have concealed the authors’ identities.

---

> > ### Comment · Reviewer_hf9n · 2023-08-29
> >
> > Dear authors, thank you for addressing my concerns. I stand by my judgement that this is a strong paper and a clear accept.

---

### Official Review · Reviewer_4J35 · 2023-07-22
**Review of Chinese Multi-Modal Dataset**

**Rating:** 9
**Confidence:** 4

**Strengths:**

This paper is well written and the methodology used to develop the resources is well explained. Significant labor was used to process the data with multiple annotations, and it with the wealth of material, it should be feasible for even groups with limited resources to perform effective research (for example. building text-only classifiers for teams that cannot effectively process speech or video should be quite easy.)

The initial experiments use a good sampling of current techniques.

**Additional Feedback:**

None.

**Clarity:**

This paper is well written and clear, with only a few minor grammatical issues (that are systemic, however, and seen throughout the paper.)

**Correctness:**

I see no obvious concerns with the construction of this dataset, and the evaluation methods and experiments all appear sound.

**Documentation:**

The documentation looks good. My only concern is question 4 of the checklist which should have been filled out for parties releasing new datasets. In particular this data set include a large collection of television content with some spanning as much as 25 seconds. While I believe that all of this content would likely fall under fair use, the authors did not clearly indicate what the license terms are for the annotation material they are providing, and did not discuss whether or not they discussed rights to distribute the television content.

**Ethics:**

As all of the material in this dataset is obtained from televisions shows, so there should be no consent or other ethical concerns.
The large annotation effort was performed by graduate students, but they were compensated for this work at an hourly rate, and we fully informed about the nature of the work as received training in preparation.
While there may be ethical risks in building systems to detect emotions, this practice is already routinely done, so I see no serious new risks posed by this work.

**Limitations:**

Sourcing content from television shows may introduce a number of biases, and it's not hard to imagine that the effects of actors portraying emotions may be either ineffective or exaggerated. This may be mitigated by the annotation process where trained annotators are marking the perceived emotions. However, it is important to acknowledge the artificiality of the entertainment medium and the potential for distributional shift that might make classifiers trained on this material less effective in other applications.

In addition, it is possible that there may be related effects concerning the coverage of topics due to both government or entertainment companies censorship. This would likely not invalidate the work here, but it is something to keep in mind. The authors do acknowledge the bias towards professionals and particular age which is a reflection of the biases in the entertainment industry.



**Opportunities For Improvement:**

There are some grammatical issues, the authors use "affection" when they mean "affect", as in affect type. Affection means feelings of warmth and liking and it's use as a name for other emotions is archaic.

Line 32: defection -> detection

The check marks in Table 1 are a little hard to distinguish

Line 188: has -> have.

**Relation To Prior Work:**

The related work section looks sound, and the table comparing the features of the previous datasets looks quite comprehensive. However, I'm not an expert in this field.

**Summary And Contributions:**

This dataset and the elaborate procedure used to create the labels should produce an invaluable and large scale resource that will enable a wide variety of research beyond the specific task created for the release. Evaluating the sentiment using video, audio, and text in context in television shows has been done before, but the granularity and depth of detail regarding the content is really unprecedented. The experiments presented in this paper are a very good starting point, and they even reach some preliminary conclusions about the importance of context for these types of tasks.

My high assessment for this paper is primarily a reflection of the care and effort that was put into designing the annotation schema and platform, and the size of the overall annotation effort. Although I do not have access to the dataset itself, if it is as described in the paper, it is really a huge step up from what has been available to date.

---

> ### Author Response · Authors · 2023-08-10
>
> $Q1$: There are some grammatical issues.
>
> $R1$: Thanks for your suggestion. We have corrected them and revised the whole manuscript in order to make it more understandable both in terms of presentation and language.
>
> $Q2$: Sourcing content from television shows may introduce a number of biases, and it's not hard to imagine that the effects of actors portraying emotions may be either ineffective or exaggerated.
>
> $R2$: Thank you for your valuable feedback on our work. We appreciate your concern regarding the potential biases that may arise when sourcing content from television shows.
>
>    But we have to say that TV shows can still serve as a valuable resource for affect detection research, due to three reasons:
>     (1) The perceived emotions of actors are not purely fabricated, but rather portrayals of genuine human experiences and reactions. Though they are likely to be more exaggerated than real-world expressions, by having NINE trained annotators label affects based on perception rather than intentions, we can collect labels that capture the essence of displayed affect.
>     (2) TV content is easy to obtain, has no privacy concerns, and can be open published and reused. Real conversation collection requires obtaining a large number of recordings or texts from actual dialogue situations. This process involves privacy concerns and legal compliance issues.
>     (3) TV show dialogues cover a wide range of language styles and affect expressions, allowing collection of richly annotated training samples.
>
> Of course, we understand the importance of addressing biases. To mitigate these concerns, we have made several efforts:
> (1) Data Diversity: We took great care in selecting a diverse range of TV series with various subjects and styles. By incorporating a wide variety of contexts, characters, and scenarios, our dataset, CMMA, aims to capture a more representative sample of human emotions in conversational settings.
> (2) Manual Annotation: The affect labels in CMMA were manually annotated by Nine well trained experts. This process helps ensure that the affects captured in the dataset are based on genuine expressions and helps minimize subjectivity and ensure accuracy.
> (3) Robust Analysis: In our experiments, we have performed rigorous analysis. We have also discussed the limitations and provide transparency about the dataset’s scope to ensure its appropriate usage (Sec. 5, Page 10).
>
> Despite the potential limitations, we firmly believe CMMA offers valuable insights into the multi-modal and multi-affection nature of human communication. Our work lays a strong foundation for further research in human affect understanding.
> We sincerely appreciate your thoughtful feedback, and we have added the detailed explanations in Appendix M in Supplementary document (Page 15).
>
> $Q3$: The authors do acknowledge the bias towards professionals and particular age which is a reflection of the biases in the entertainment industry.
>
> $R3$: Your questions and concerns are valuable to us. We have discussed these biases and strived to maintain transparency and diversity regarding the dataset. We hope that the research community understands these limitations and takes them into careful consideration when using the dataset (Page 10).
>
> $Q4$: The authors did not clearly indicate what the license terms are for the annotation material they are providing, and did not discuss whether or not they discussed rights to distribute the television content.
>
> $R4$: Thank you for raising this important issue. Regarding the copyright concerns, we have explicitly argued that CMMA is made available for research purposes only and added the detailed licensing statements (CC BY-NC 4.0 License) on Github and Appendix 1.3. (Lines 37-41, Page 2).

---

> > ### Comment · Reviewer_4J35 · 2023-08-29
> >
> > All of my concerns (which were minor) have been adequately addressed. I stand by my rating which was higher than the other reviewers. Thanks for this valuable and labor intensive contribution.

---

### Official Review · Reviewer_GqNB · 2023-07-23
**Review of CMMA**

**Rating:** 7
**Confidence:** 5
**Correctness:** The paper is sound
**Clarity:** The paper is clear in its descriptions

**Strengths:**

As mentioned above in summary and contributions

**Additional Feedback:**

Nothing particular

**Documentation:**

Sufficient details are given.

**Ethics:**

No specific ethics concerns

**Limitations:**

The limitations are well described

**Opportunities For Improvement:**

Linguistic insight and case studies are sadly missing!

**Relation To Prior Work:**

Table 1 is a very good depiction of placing the dataset in relation to other similar datasets.

**Summary And Contributions:**

The paper gives the first Chinese Multi-modal Multi-Affection conversation (CMMA) dataset, which contains 3,000 multi-party conversations and 21,795 multi-modal utterances collected from various styles of TV-series. CMMA contains a wide variety of affection labels, including sentiment, emotion, sarcasm and humor, as well as the novel inter-correlations values between certain pairs of tasks. Moreover, it provides the topic and speaker information in conversations, which promotes better modeling of conversational context.

---

> ### Author Response · Authors · 2023-08-10
> **Response to the question of "Linguistic insight and case studies"**
>
> $Q1$: Linguistic insight and case studies are sadly missing!
>
> $R1$: We sincerely appreciate your constructive feedback. We acknowledge the importance of providing linguistic insight and case studies to further enrich the analysis.
>
> (1) Case study: Actually, we have already made case study in the Appendix H (Error Analysis, Pages, 11-13) of Supplementary Document. In this section, we have showed 16 multi-modal cases (utterance plus image) to provide a detailed examination of particular samples, shedding light on the complexities and challenges involved in multi-affection analysis tasks. We have also presented the distribution of misclassification cases for sentiment, emotion, sarcasm and humor to highlight the challenges and limitations.
>
> (2) Linguistic insight: We acknowledge the importance of linguistic insight in understanding the multi-modal and multi-affection nature of human communication. Due to the limited time, we have only analyzed the usage of the emotional terms and the distribution of rhetorical devices (in Appendix L, Page 14-15). More thorough linguistic analysis will be introduced in the new version.

---

### Decision · Program_Chairs · 2023-09-22

**Decision:**

Accept (Poster)

**Comment:**

This paper introduces a new dataset of Chinese TV series, annotated with a variety of affection categories. Reviewers agree on the usefulness of the dataset, however, multiple reviewers, as well as the authors themselves raise concerns about the biased data, due to compilation of acted speech. I agree with this concern, and suggest extending this data with natural (i.e., not acted) conversational speech as well.